# Corollary discharge enables proprioception from lateral line sensory feedback

**Dimitri A. Skandalis**[1,2]*, **Elias T. Lunsford**[1], **James C. Liao**[1]*

**1** Department of Biology & Whitney Laboratory for Marine Bioscience, University of Florida, St. Augustine, Florida, United States of America, **2** Department of Psychological & Brain Sciences, Johns Hopkins University, Baltimore, Maryland, United States of America

* da.skandalis@gmail.com (DAS); jliao@whitney.ufl.edu (JCL)

**Data Availability Statement:** Data and modelling code have been deposited in the online repository available at https://doi.org/10.6084/m9.figshare.13034012.v1.

## Abstract

Animals modulate sensory processing in concert with motor actions. Parallel copies of motor signals, called corollary discharge (CD), prepare the nervous system to process the mixture of externally and self-generated (reafferent) feedback that arises during locomotion. Commonly, CD in the peripheral nervous system cancels reafference to protect sensors and the central nervous system from being fatigued and overwhelmed by self-generated feedback. However, cancellation also limits the feedback that contributes to an animal's awareness of its body position and motion within the environment, the sense of proprioception. We propose that, rather than cancellation, CD to the fish lateral line organ restructures reafference to maximize proprioceptive information content. Fishes' undulatory body motions induce reafferent feedback that can encode the body's instantaneous configuration with respect to fluid flows. We combined experimental and computational analyses of swimming biomechanics and hair cell physiology to develop a neuromechanical model of how fish can track peak body curvature, a key signature of axial undulatory locomotion. Without CD, this computation would be challenged by sensory adaptation, typified by decaying sensitivity and phase distortions with respect to an input stimulus. We find that CD interacts synergistically with sensor polarization to sharpen sensitivity along sensors' preferred axes. The sharpening of sensitivity regulates spiking to a narrow interval coinciding with peak reafferent stimulation, which prevents adaptation and homogenizes the otherwise variable sensor output. Our integrative model reveals a vital role of CD for ensuring precise proprioceptive feedback during undulatory locomotion, which we term external proprioception.

## Introduction

Maneuvering through the environment requires an awareness of the body and its movements, a sense called proprioception. Proprioception enables smooth actions like reaching and grasping for a doorknob in the dark or rapid righting responses to perturbing effects like walking on an unstable surface [1]. Importantly, proprioceptive feedback is not simply a readout of body and limb positions and angles but a dynamic sense that depends on motor context. For example, the gain of type 1a (muscle spindle) afferent fibers is modulated according to gait cycle

**Funding:** This research was funded by National Institutes of Health grant R01 DC010809 (https://www.nih.gov/) and National Science Foundation grants IOS1257150 and IOS1856237 (https://www.nsf.gov/) to JCL. The funders had no role in study design, data collection and analysis, decision to publish, or preparation of the manuscript.

**Competing interests:** The authors have declared that no competing interests exist.

**Abbreviations:** CD, corollary discharge; GAM, generalized additive model; GLM, generalized linear model; ISI, interspike interval; nAChR, nicotinic acetylcholine receptor; PSTH, peristimulus time histogram; VR, ventral root.

phase [1,2]. The neural signals that modulate sensor gain during motor activity are called corollary discharge (CD) and are widespread across the animal kingdom [3,4]. CD is a signal transmitted in parallel to motor commands that modulates sensor feedback according to the predicted magnitude of self-induced feedback (reafference) [3,4]. Often, sensor gain is reduced because sensors respond equally to self-induced as to environmental stimuli (exafference). Consequently, vigorous body movements will be translated to a large volume of reafference that risks overwhelming the nervous system and masking both the proprioceptive signal and environmental stimuli. How CD modulates proprioception should be especially important for locomotion in fluids. The need to operate with millisecond precision [5–8] may be challenged by nonlinearities introduced by the fluid itself and by heterogeneous sensor gain and adaptation rates within sensor and afferent populations [1,7,9–11]. System variability fundamentally limits the accuracy of processing in the central nervous system [7,8]. We examine the role of CD in regulating sensor adaptation and heterogeneity to maximize spikes' proprioceptive information content in an aquatic environment. We develop this role in the context of fishes' lateral line system, the proprioceptive capacity of which has long been suspected but difficult to verify [12,13].

To fully understand the contribution of proprioceptive feedback within motor control, we must know which motor and mechanical signals are being transduced by sensors. This is an especially challenging problem for continuum joints in which shape varies continuously [14], such as that of the fish body during undulatory locomotion. While swimming, the fish's body curvature varies from head to tail according to the phase of the motor cycle (Fig 1A) [15]. A fish must therefore sense and control body angles over its axial length through successive muscle layers that may span up to 10 vertebrae, rather than one or a few dedicated muscle joints [16]. In robotics, feedback from continuum joints is typically provided by sensing changes in length and curvature [14]. However, it has recently been demonstrated that stable swimming rhythms can be generated through sensing of hydrodynamic pressures on the body [17]. Fish possess few, if any, length sensors like muscle spindles [18,19], but can sense body curvature in the spinal cord and through stretching of the skin [20–22]. There is a long-standing question of whether fish can also receive proprioceptive feedback through flow sensing in the lateral line (Fig 1) (reviewed by [12,13]). The lateral line transduces fluid motions within the boundary layer through the bending of neuromast cupulae, depolarization of hair cells, and, ultimately, the transmission to the brain of afferent fiber spike activity [10,13,23–25] (henceforth collectively lateral line sensors). Basic rhythmic motions like respiration are well known to be represented by spiking in the brain [26,27]. In swimming sea lamprey, the sister clade of jawed fishes and a comparative model of vertebrate evolution, unmodulated lateral line feedback in the brain robustly encodes periodic body motions [12]. Ablation of the sea lamprey lateral line results in increasingly distorted kinematics as swim speeds increase, pointing to a need for body awareness as motor effort increases [12]. In jawed fishes, ablation of the lateral line likewise compromises efficient navigation of turbulent flows [28,29], and anterior lateral line feedback is necessary for coordinating head and trunk motions to optimize sensing and respiration during forward swimming [30]. These roles highlight the specific need for the animal to be aware of its body configuration within the invisible eddies of the fluid environment [31–33]. Because this feedback depends on sensing body motions through coupling with the boundary layer, we refer to it as a sense of external proprioception.

The biophysics of lateral line sensors might limit the potential for proprioception. As a whole, lateral line sensors are highly variable, which is thought to benefit exteroception (sensing of exafference) through spatial integration and by imparting a wide sensitivity range and frequency bandwidth [9,10,23,34]. However, because the lateral line is a near-field sensor, reafference will typically be a greater stimulus than exafference [12,13,35]. Highly sensitive and

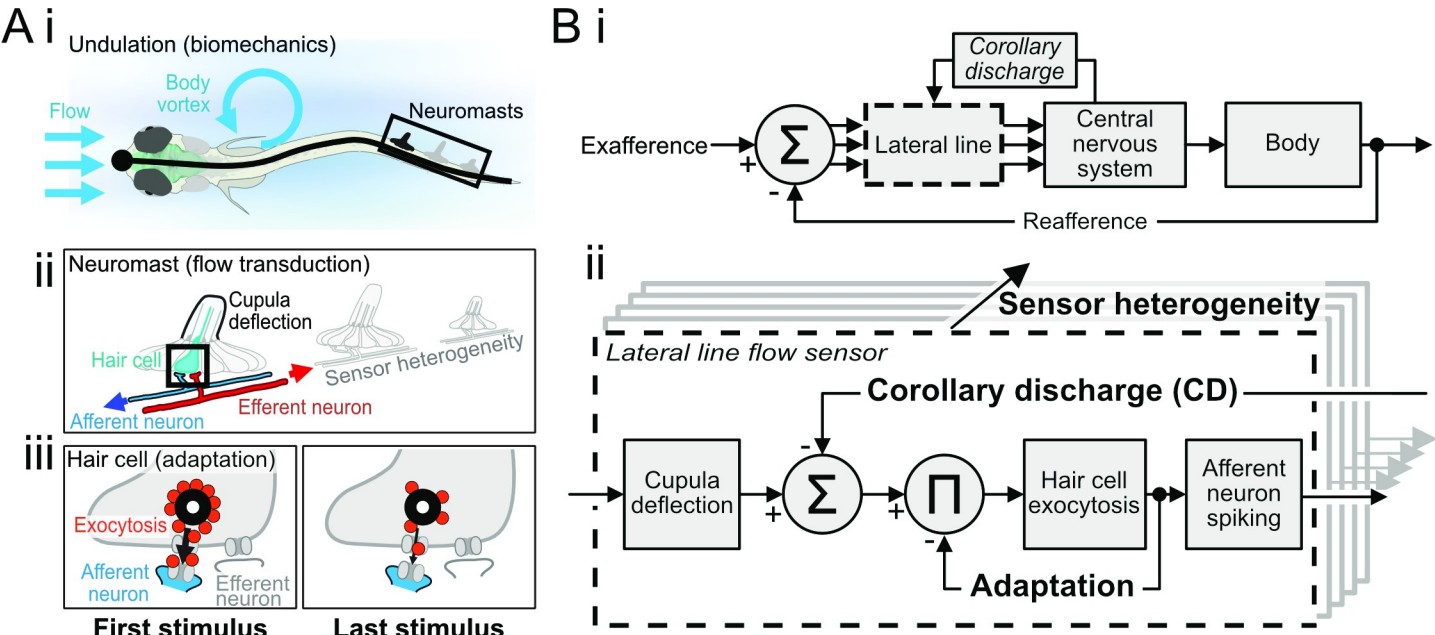

**Fig 1. A systematic approach for cellular-to-organismal integration in swimming fish. (A)** (i) Fish sense water flows that originate in the environment (exafference) or are self-induced by undulatory locomotor mechanics (reafference). Both sources generate alternating currents across a wide band of frequencies, which are transduced by neuromasts along the body length. (ii) Afferent neuron signaling of neuromast deflections is mediated by receptor potentials in hair cells, as modulated by inhibitory efferent neuron activity. (iii) Depolarization of hair cells results in vesicle exocytosis onto postsynaptic afferent neuron boutons. Sustained neuromast deflection results in adaptation, here depicted as synaptic depression. **(B)** Hypothetical architecture of closed-loop motor control in fishes. (i) Many individual sensors in the lateral line transduce the summed exafferent and reafferent inputs. Each sensor transmits a filtered signal to the central nervous system. Brain computations extract key flow features that guide motor actions. Motor signals drive behavior, and motor signal copies called CD modulate lateral line feedback in anticipation of self-induced feedback from those behaviors. (ii) Elaboration of the sensor block to understand the effects of CD modulation on lateral line feedback. Cupula deflection drives vesicle exocytosis from hair cells, which leads to synaptic depression and adaptation of afferent neuron spike rates. Inhibition by CD reduces exocytosis and therefore afferent spike rates, but also prevents adaptation. This is replicated throughout the lateral line, so the interactions between adaptation, CD, and sensor heterogeneity are central to understanding the nature of feedback during swim bouts. CD, corollary discharge.

heterogeneous sensors that are beneficial in a stationary fish thus risk becoming detrimental in a swimming fish if they transmit an overwhelming volume of feedback. The salience of both proprioceptive and exteroceptive signals will be reduced if significant computation is required to discriminate reafference from exafference and then to distinguish specific features of the reafference. Behavioral decisions would therefore be more difficult to execute while swimming, which could lead to the animal mistaking vigorous reafference for an impending attack or to it missing a real attack altogether [3,36,37]. Besides behavioral selection, hair cell systems like the lateral line exhibit high stimulus gain (up to nanometer sensitivity [10,38]) combined with finite vesicle populations [11,39]. Sustained reafference fatigues sensors at different rates [10], reducing sensitivity and distorting the timing of neural signals [40–42]; in the extreme, over-stimulated sensory circuitry may lead to excitotoxic cell death [43]. Proprioceptive capacities of the lateral line for controlling the body as a continuum joint would therefore seem to be intrinsically limited by time-varying changes in the gain and temporal faithfulness to the reafference.

The utility of peripheral cancellation of reafference in other animals has led to the view that CD to the lateral line likewise cancels reafference [3]. This view is based on the principle that cancelling reafference could increase the robustness of behaviors that depend on tracking hydrodynamic stimuli through the lateral line, like schooling, prey detection, and rheotaxis [29,44–47]. Cholinergic projections from the hindbrain hyperpolarize hair cells and reduce spontaneous and evoked afferent spike rates (efferent neurons in Fig 1Aii) [13,48,49]. The CD

is strictly activated during swim bouts [35,48,49], and laser ablation of the cholinergic cell bodies prevents inhibition [48]. Cancellation could be produced if the strength of sensor inhibition induced by the CD matches the magnitude of reafference (Fig 1B). Indeed, hair cells that are polarized to respond to the anterograde (rostrocaudal) currents encountered during forward swimming are the most heavily suppressed [49]. Nonetheless, the CD-induced hyperpolarization is insufficient to completely suppress spontaneous spiking [48–50] let alone evoked activity [49,50]. Reafferent spikes will likely always be transmitted to the brain, depending on motor effort [51,52], which is consistent with the suggestion that CD represents a sensory goal rather than sensory prediction [53]. In this way, the spikes can encode the mismatch between goal and execution, such that the resulting error signal helps train the sensorimotor loop (Fig 1B). This implies that spikes transmitted despite CD inhibition do in fact carry information to the central nervous system, even in the absence of exafference. The value of these spikes is that reafference is the error signal on which motor learning and sensor planning are based (Fig 1B) [3,27,54,55]. Elimination of reafferent feedback in the phylogenetically related auditory system, as occurs in the hearing impaired, in fact compromises voice control [56]. We argue that reafference from the lateral line likewise encodes feedback that is essential for guiding motor patterns, including the animal's relative orientation, speed, and acceleration with respect to the flow [29,57,58].

We hypothesized that CD serves a critical role in regulating sensor adaptation from heterogeneous sensors (Fig 1Bii). Hair cell activity is autoregulated by activity-dependent pre- and postsynaptic mechanisms, including synaptic depression [10,11,39,59,60], which is graphically represented by vesicle depletion in Fig 1Aiii and is schematically represented by negative feedback in Fig 1Bii. We predicted that because CD acts upstream of hair cell signaling to suppress evoked vesicle release, the CD also suppresses the nonlinear effects of adaptation. Our goals in this work were 2-fold. First, we develop a neuromechanical model of proprioceptive signals and their transduction in the fish body continuum joint. Second, we build on previous examinations of the independent effects of heterogeneity, adaptation, and CD in the lateral line [10,49] by examining the interactions between these factors. These interactions are critical for understanding how sensory processing is modulated during the behavioral shifts between rest and motor activity. The proposed proprioceptive function of the lateral line is enabled by the central role of CD in regulating the magnitude and temporal characteristics of feedback from populations of heterogeneous sensors. The capacity of CD to regulate spike temporal codes as well as rate may have broad significance throughout the nervous system.

## Results

### A biomechanical theory of external proprioception

We first consider how feedback may signal the current configuration of a body. This is a challenging problem in a continuum joint like the fish body because of the high degrees of freedom and deformations under applied loads. We note that in the absence of CD, reafference encodes periodicity of fishes' body motions through spiking in the lateral line [12,13,26,27,35]. On the basis that lateral line ablation consistently leads to dysregulation of the mechanical body wave [12,28], sensing the body wave itself can be the proprioceptive signal. Specifically, we propose that fish detect the peak of the body wave as it passes each neuromast (Fig 2A). Peak detection within a noisy signal is fundamentally challenging, which can be exacerbated by highly heterogeneous sensors with nonlinear response characteristics. When subject to stimulation, the responses of hair cells and afferent neurons readily adapt [10,39,59,60] (Fig 1). A major cause is vesicle depletion, resulting in decreased sensitivity to stimuli and distortions of afferent spike timing [40,42]. Ostensibly, nonlinear operations in the brain might counter this effect.

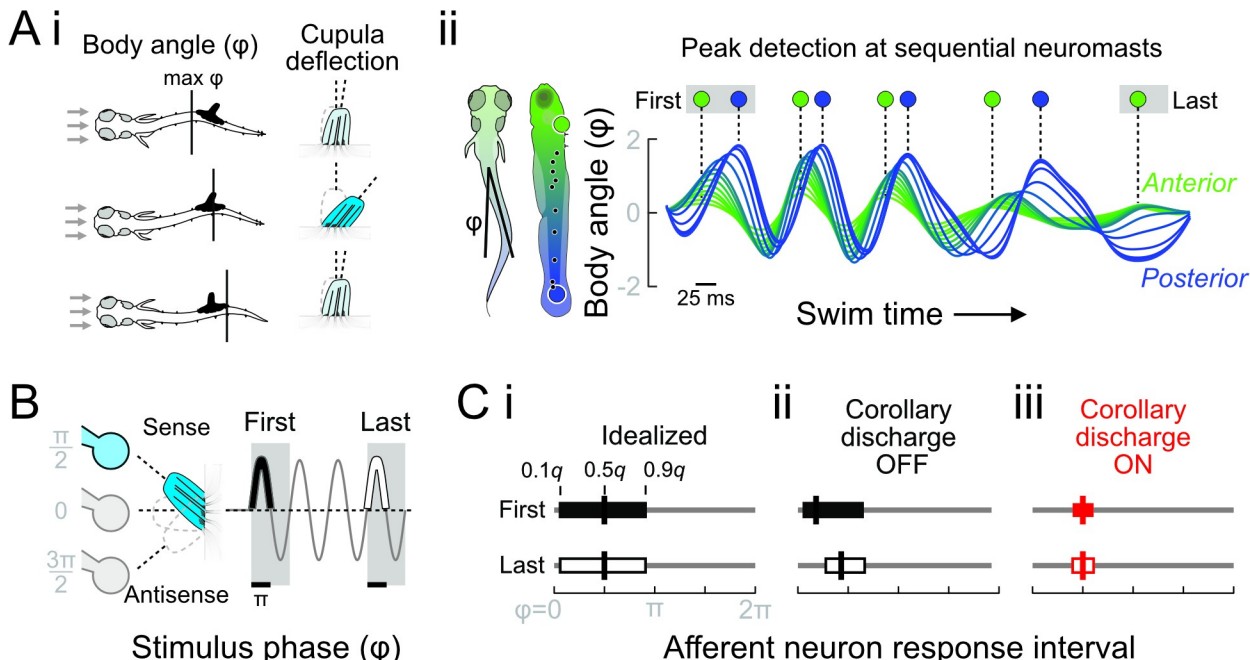

**Fig 2. Effects of undulation, adaptation, and CD on signaling of reafference in the lateral line. (A)** (i) Larval zebrafish swimming is characterized by body undulations that induce periodic reafference in the lateral line. The instantaneous body angle at each neuromast (φ) depends on body wavelength and frequency. In the quasi-steady approximation, the body wave passing a neuromast dictates the timing of maximum cupular deflection (denoted by intense blue). (ii) The timing of the body wave peak from the first to last tail beats is shown at the locations of the most rostral (green) and most caudal (blue) neuromasts. **(B)** Neuromasts were stimulated by a dipole piezo stimulus to examine afferent neural responses from the first to last stimulus period. In hair cells' sense direction, the peak of feedback is expected to coincide with maximum stimulation at phase π/2 (denoted by intense blue). **(C)** Polarized hair cells have a maximum signaling phase of a half stimulus cycle. (i) An idealized sensor maintains a proportional output between deflection and output throughout the stimulus bout. The response interval is therefore constant, as reported by the quantiles (q) of afferent spike phase with respect to the stimulus (φ). Thick vertical lines denote the median spike phase. (ii) In real sensors, we observe adaptation from the first to last stimulus intervals. We predict this leads to decreased sensitivity and thus distorted response intervals and increased VS. Peak reafference precedes π/2 and changes over time, so cannot reliably signal body wave progression. (iii) CD (ON) reduces hair cell sensitivity during motor bouts, so hair cells provide feedback only to the strongest cupular deflections coinciding with the passage of the body wave. CD thus restructures afference to reliably signal motor phase throughout the motor bout. CD, corollary discharge; VS, vector strength.

However, the lateral line is composed of many highly variable elements. Factors like variation in cupula length and the number and maturity of hair cells [23,34,61] underlie variation in adaptation rates even within a single neuromast [10,23,34]. Adaptation-associated distortions thus occur not just once but are repeated at every synapse between hair cells and afferent neurons (sensor heterogeneity in Fig 1). The combinatorial complexity of this variability would pose an enormous computational challenge for the brain. The capacity to learn these heterogeneous nonlinearities is moreover restricted by continuous turnover and maturation of hair cells throughout a fish's life [61], so that the nature of reafference is unlikely to be constant. We hypothesized that CD is a key element of the lateral line circuit that modulates hair cell gain to prevent adaptation and its associated nonlinearities.

We propose that fish determine the joint's configuration by sensing the location of the peak of the body wave. In the quasi-steady approximation, the curvature of the body causes flow acceleration like around a bluff body. Therefore, as the peak passes each cupula, the body curvature will cause a transient elevation in flow, resulting in a maximum of cupula deflection (Fig 2Aii). Accurate feedback on peak timing between sequential neuromasts can be used to calculate parameters of the body wave, adding to the diversity of flow calculations performed in fish brains [32,62,63]. Ultimately, the computations that occur in the brain will depend on

how the peripheral nervous system handles nonlinearities in the lateral line feedback. We examined 4 parameters of lateral line afferent neuron spiking to study the conditions in which lateral line feedback can provide faithful feedback on the progression of the body wave. We first examined the number of evoked spikes per stimulus, a basic measure of feedback. However, the number of spikes alone is unlikely to provide the capacity for peak detection, since it cannot discriminate between the rising edge, peak, and falling edge of the stimulus. Motor systems in fluid environments can depend on spike timing with millisecond precision [5], and the lateral line is thought to encode stimuli with similar precision [8]. We therefore examined spike timing by the location and variability of spike phases with respect to the stimulus (0 to $2\pi$; simplified sinusoidal stimulus depicted in Fig 2B). The median spike phase (0.5 quantile, $q$) was used to determine spikes' tendencies to lead the stimulus (median $< \pi/2$) or to be distributed around its peak (median $= \pi/2$). Spikes with highly variable timing convey less information about the stimulus. The response variability was examined by the width of the phase response interval, which is related to the vector strength. The response interval width was determined by the timings of early and late arriving spikes (0.1 and 0.9 $q$; the earliest spikes are also called the spike latency, e.g., [42]). Both the response interval width defined by the quantiles and the commonly reported vector strength are functions of the circular variance [64]. A narrow interval has high vector strength and reflects low variance in spike timing, meaning that spikes are reliably evoked at a certain phase of the stimulus (phase locked). Finally, we report the spike gain, which reflects the salience of a distinct feature of the stimulus, such as its maximum. The spike gain was calculated by the product of spike count and the vector strength [24]. This allowed us to discriminate between scenarios in which multiple evoked spikes were uniformly distributed over the stimulus cycle (low gain) and a single, highly phase-locked spike (high gain).

We first consider the output of a hypothetical, idealized sensor that linearly transduces all cupular bending to an output spike rate code (Fig 2Ci). In the idealized sensor, the response interval encompasses bending in the sense direction [10,49,65]. The median phase coincides with maximum cupular deflection at $\pi/2$ because this is the peak of stimulation (stimulus phase depicted in Fig 2B). Although the spike rate encodes the peak location, the sensor's wide response interval (low vector strength) means that individual spikes within single swim bouts occur probabilistically at any phase of bending in the sense direction (Fig 2B). Accordingly, the sensor has low gain because individual spikes within any single cycle cannot reliably signal peak deflection. To provide high-quality proprioceptive feedback, the idealized sensor should therefore also have a narrow response interval.

In contrast to the idealized sensor, real hair cells exhibit substantial adaptation that induces nonlinearities in afferent feedback [10,42,60]. Hair cells' vesicles are primed to rapidly release in response to stimulation, resulting in an initially strong and phase leading response. Subsequently, reduced vesicle availability means that a larger cupular deflection angle is required to evoke the same response. For the same stimulus, the response interval therefore becomes delayed and narrowed (Fig 2Cii). The quality of the proprioceptive feedback depends on the location of the response interval with respect to the stimulus peak. Moreover, because lateral line sensors are highly heterogeneous [10,66], the individual responses of adapted sensors will in general be unpredictable. Adaptation is an auto feedback response to evoked stimulation and therefore minimizing evoked responses by CD will likewise minimize adaptation (Fig 1B). While swimming, CD carried by cholinergic efferent neurons hyperpolarizes hair cells so that a larger cupular deflection and depolarizing current is required to achieve the same evoked spike rate (Figs 1B and 2Ciii) [35,48,49,59]. When the cupula is deflected from rest along its preferred axis, spiking will be reliably evoked only when the deflection passes the higher threshold set by CD, i.e., the hyperpolarization offset. This results in a delay of early spikes.

During the falling phase of the stimulus after the peak, spiking can be evoked until the cupular deflection again passes the CD-induced threshold. As the level of hyperpolarization increases, the delay of early spikes and advance of late spikes will shrink the response interval, as depicted in Fig 2Ciii. Eventually, sensory responses will be restricted to only the strongest phase of the input stimulus and thus will occur in a narrow band (response interval) centered on $\pi/2$. The change in threshold due to CD is therefore equivalent to sharpening lateral line sensors' polarization by rejecting subthreshold stimuli. Even though the CD may reduce the number of spikes per stimulus, their gain is high because the fish can reliably detect the location of the peak from single spikes (note that in our formulation, the gain is always equal to or less than the spike count). In principle, adaptation could have the same effect if it strongly reduces the spike count and delays and narrows the response interval. As noted above, this would be unpredictable given lateral line sensors' heterogeneity. Importantly though, the function of CD to inhibit vesicle release will protect the vesicle pool from depletion so that it is primed to robustly respond to flow perturbations after the swim. On the other hand, as discussed below, excessive inhibition by the CD can also introduce post-swim artifacts. We therefore paid special attention to the effect of CD, stimulation, and sensor heterogeneity on spike properties through the swim bout and into the post-swim period. In sum, the location (median phase) and gain (product of spike count and vector strength) of evoked responses are key to understanding their ability to facilitate proprioceptive content. Changes in these properties over the course of the swim or due to CD will reveal how sensor nonlinearities are regulated.

## Interactions between adaptation and corollary discharge

We studied fictive reafference through a sinusoidal stimulus (Fig 2B). Frequency responses are a major component of sensory systems and ultimately of feedback loops [27,67], and, therefore, our interpretation of the significance of adaptation, CD, and proprioceptive information content may be greatly influenced by the choice of stimulus frequency. Typical tail beat frequencies of freely swimming zebrafish larvae lie between 20 and 25 Hz [51]. Many studies of larval zebrafish lateral line function, including of efferent modulation, have examined frequencies much lower than the predicted reafference (e.g., <1 Hz stimulation [10,42,49]). We stimulated at 5, 20, and 40 Hz, representing stimulation in the lower quantiles, near the peak, and in the upper quantiles of the typical reafference frequency distribution. Where previous studies have examined a wider range of frequencies, including frequencies relevant to reafference, responses have often been averaged over durations >1 second [24] and up to 15 minutes [68]. Larval zebrafish swim bouts are short, ranging from 3 to 15 tail beats (60 to 300 ms at 20 Hz) depending on the behavior [51,52]. Averaging over long periods may therefore obscure rapid adaptation dynamics relevant to fish behavior. We stimulated for 1 second, the upper time limit of swim bouts recorded here (S1 Fig), and then statistically quantified rapid changes in sensory dynamics from the first to last stimulus intervals.

We stimulated single neuromasts while recording swim behavior and spikes from single afferent neurons within the posterior lateral line ganglion (loose patch configuration; Fig 3Ai). Spikes were labeled as evoked when they occurred within a stimulus interval or as spontaneous if not. Evoked activity displayed the adaptation characteristic of hair cell systems [10,11,60,69] and a subsequent refractory period before returning to baseline spontaneous spike rates (Fig 3Aii, S2 Fig). Evoked spikes were therefore additionally labeled according to their stimulus interval. In fish, the CD to the lateral line is strictly active during swim bouts [35,49], so each spike was also labeled as CD ON or OFF depending on whether it coincided with motor activity. Changes in spike rates due to CD were quantified by the ratios of activities ($R$ = CD ON/CD OFF) during spontaneous ($R_S$) and evoked ($R_E$) periods (Fig 3A). On average, $R_S$ and $R_E$

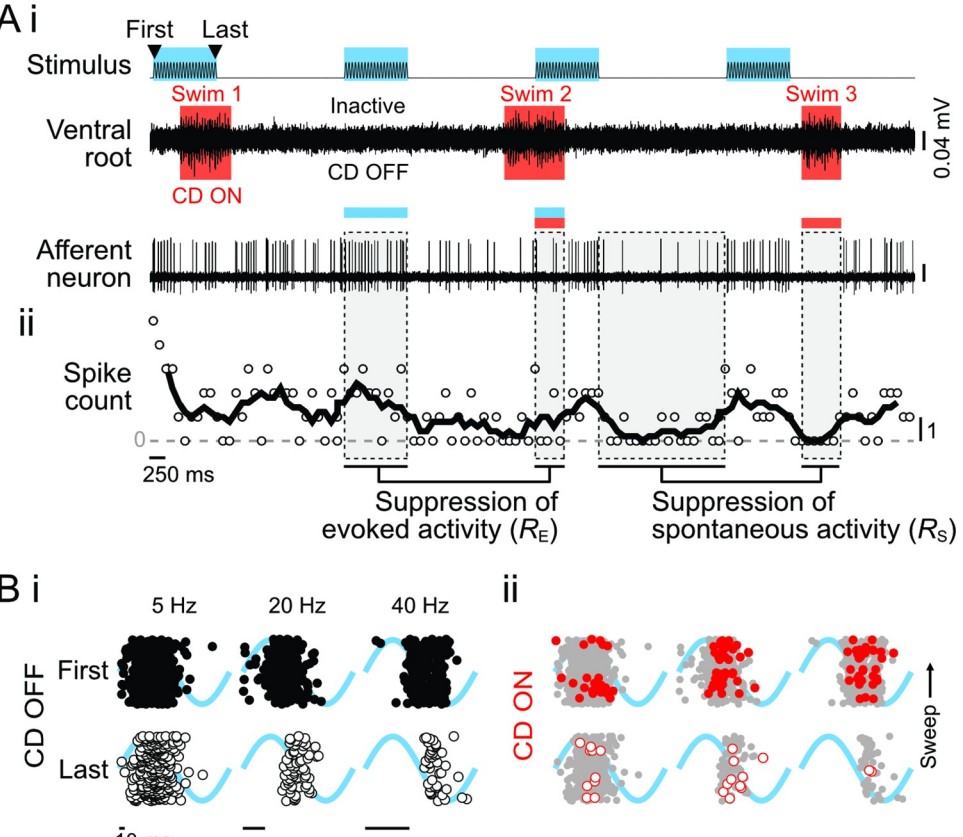

**Fig 3. Dynamics of afferent neuron spike activity in response to neuromast deflection depend on CD. (A)** (i) Experiments were performed during open-loop stimulation (first to last interval), during which fish swam spontaneously (motor activity recorded through ventral motor root). The CD was ON during swim bouts and OFF while the fish was inactive. Afferent neuron spiking depended on the combination of stimulation and motor activity, as shown by the moving average of spike count (ii). Individual spikes were classified as evoked (E) when the stimulus was present or spontaneous (S) otherwise. The effect of CD on evoked and spontaneous activity was quantified by the ratio of CD ON/CD OFF, respectively, $R_E$ and $R_S$. **(B)** Raster plot of evoked afferent spike phases during the first and last periods of 5, 20, and 40 Hz stimulation. Stimulus phase is underlaid in blue. (i) In this afferent neuron, when CD OFF the afferent neuron exhibited a strong response to the first stimulus, but a delayed and weaker response to the final stimulus. (ii) In CD ON, there was a reduction of both the number of spikes and the response interval with respect to CD OFF (replicated in gray). In each plot, sweeps progress from bottom to top. All data from one representative individual. Gaps between sweeps arise when the fish ceases swimming due to open-loop stimulation [70]. The data and code underlying this figure may be found at DOI: 10.6084/m9.figshare.13034012. CD, corollary discharge.

were approximately 50% ($p < 0.001$, S1 Fig; 5,859 total swims), comparable to previous studies [48,49]. (This average slightly differs from that obtained while accounting for response type heterogeneity, as in the models described below.)

Both adaptation and CD affected spike phase along with spike rate, as demonstrated by raster plots of the first and last stimulus intervals across the 3 stimulus frequencies (Fig 3B). At 5 Hz in the absence of CD, there was relatively little change in either the number or phase of spike responses (Fig 3Bi), whereas at 20 and 40 Hz, there was a pronounced delay of the earliest and median spike phases but relatively little effect on later spikes. CD likewise affected spike phase at all frequencies (Fig 3Bii) and most evidently during 5 Hz stimulation in which the response interval was greatly narrowed. The raster plot of CD ON at 40 Hz reveals the potential confound of spike rate suppression on spike phase estimation. Due to the open-loop experimental design, the probability of observing a spike during a swim bout depends on the effects

of adaptation, the CD strength, and the probability of a swim. Strong inhibition in a fish with infrequent swimming could result in infrequent spiking, particularly in later stimulus intervals. Our statistical analysis accounted for this by averaging the effects of CD across all fish and all intervals, i.e., we calculated an intercept difference that represented the average influence of CD on responses across all intervals. This is appropriate because for any given interval, the responses have been averaged over all phases of the swim bouts, so for this analysis, we cannot resolve nonlinear effects like a greater impact of CD on the first interval than on the last.

We quantified the effects of adaptation and CD over the stimulus period through population average activities (20 Hz population averages shown in Fig 4A and 4B). Over the course of stimulation, we observed a continuous decline in spike count (Fig 4A), along with a delay and slight narrowing of the response interval (phase quantiles in Fig 4Aii, vector strength in Fig 4Aiii). A small increase in vector strength did not offset the decline in spike count, and so gain likewise declined (Fig 4Aiv). Like adaptation, CD ON reduced spike number (Fig 4Bi) and delayed the timing of early and median spikes (Fig 4Bii). However, CD ON advanced rather than delayed the timing of late spikes (Fig 4Bii). The consequently pronounced narrowing of the response interval and increase in vector strength (Fig 4Biii) partly reduced the difference in stimulus gain between CD ON and CD OFF (Fig 4Biv). Adaptation and CD are thus similar in terms of spike rate and the timing of early spikes (spike latency) but have divergent impacts on late arriving spikes and therefore the overall distribution of spike phases. These contrasting effects may substantially affect how lateral line feedback can proprioceptively encode body configuration.

We compared the relative effects of adaptation and CD by expressing each as a ratio (Last/First and CD ON/CD OFF). The effects of adaptation and CD were equal at 5 Hz, but adaptation had relatively larger effects at 20 and 40 Hz (Fig 4Ci). Adaptation induced phase delays that shifted response intervals at 20 and 40 Hz, whereas CD narrowed response intervals at all frequencies (Fig 4Cii). There was a trend of increasing vector strength due to adaptation with increasing stimulus frequency, but the changes were small and not statistically identifiable. On the other hand, CD significantly narrowed the response intervals and increased vector strength at all frequencies (Fig 4Cii and 4Ciii). These results revealed that adaptation had a greater effect on the location (i.e., median) of the response interval than on its width, whereas CD affected both. Consequently, although CD reduced spike rates, the actual impact on response gain was much less than that of adaptation (Fig 4Civ). This is most evident in the relatively small changes in response to 20 Hz stimulation, the frequency closest to typical larval zebrafish tail beat frequencies and thus reafference.

The effects of CD on spiking properties are dynamic over the course of the swim bout [49,50]. Following swim bouts that terminated within a stimulus interval, we typically observed an immediate restoration of sustained evoked spike rates (e.g., second swim bout in Fig 3A) or even bursts of evoked spikes that were otherwise seen only at the start of the stimulus presentation (examples highlighted in S3 Fig). The rapid restoration of evoked spiking suggested that CD protects sensors from adaptation and from being desensitized to stimuli that occur in the post-swim period. On the other hand, we previously found a persistent suppression of spontaneous spiking after the swim that could reduce sensitivity [48]. To assess this possibility over the wide range of swim durations, we normalized all swim bouts and an equal post swim period (see S3 Fig) to unit duration. For evoked activities, we analyzed only those swims falling entirely within or outside a stimulus presentation. Spike count rose steadily within a swim bout to a peak in the post-swim period (Fig 5A), in stark contrast to the post-stimulus depression otherwise observed (S2 and S3 Figs). Spontaneous spike rates behaved similarly ($p < 0.001$; Fig 5A), which is consistent with charge accumulation and subsequent rebound activity observed when inhibitory efferent fibers are electrically stimulated [71]. The fact that

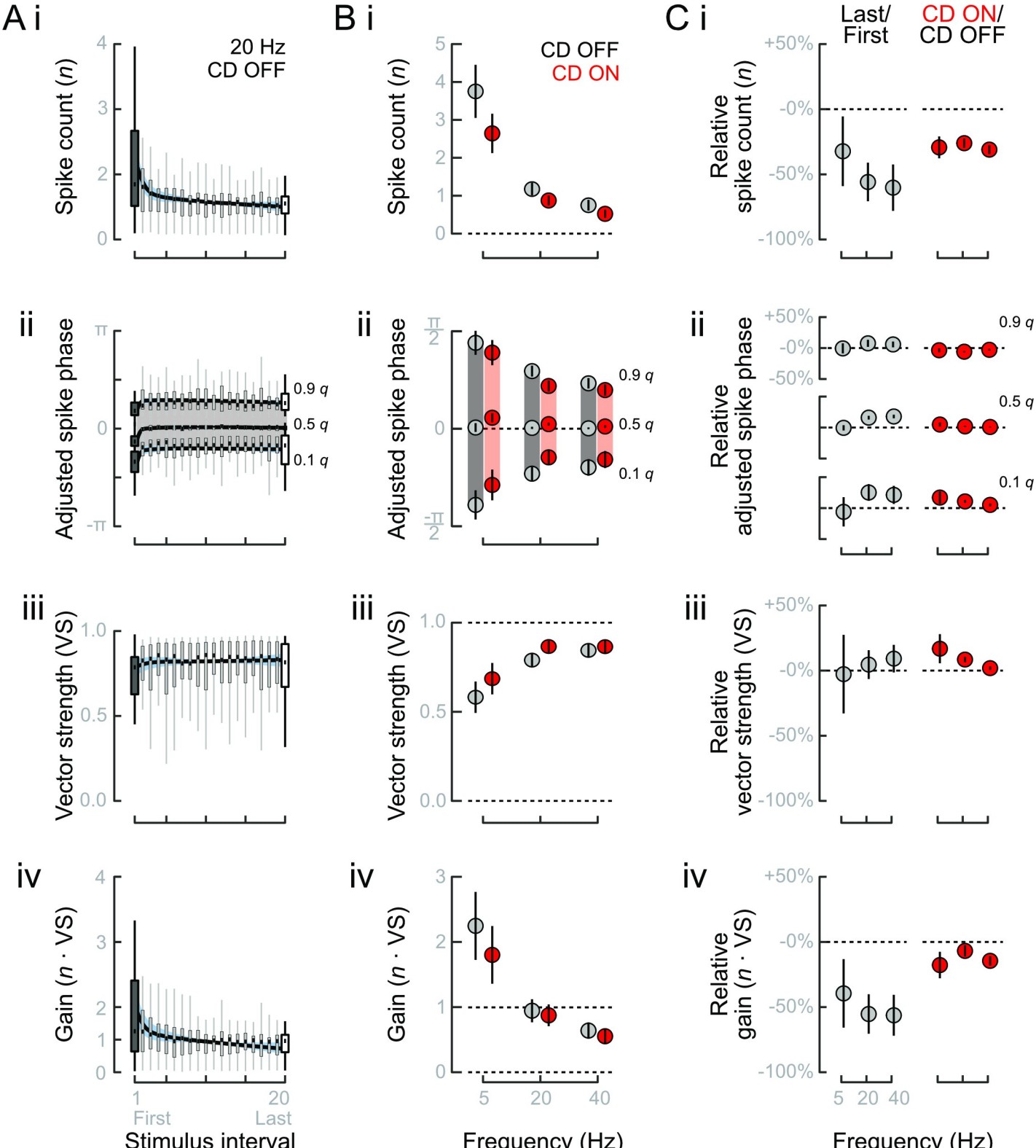

**Fig 4. Mean effects of adaptation and CD on evoked afferent fiber spike responses.** (A) Average time course of responses to 20 Hz stimulus with CD OFF. Properties of the evoked spike responses were examined through (i) the per-stimulus spike count; (ii) the response interval assessed through the 0.1, 0.5, and 0.9 quantiles of spike phases; (iii) the VS; and (iv) the gain. Spike phases were adjusted by subtracting each neuron's median phase at each frequency. (B) Mean evoked responses when CD

OFF or CD ON at each frequency. Increasing frequency and CD ON (i) reduced per-stimulus spike count, (ii) narrowed response intervals, (iii) increased VS, and (iv) reduced gain. **(C)** Relative impacts of adaptation and CD were assessed by comparing Last/First to CD ON/CD OFF. (i) Changes in spike count due to adaptation and CD were similar at 5 Hz but relatively larger at 20 and 40 Hz. (ii) At 20 and 40 Hz, adaptation resulted in delayed response intervals. Conversely, CD at all frequencies resulted in narrower intervals. (iii) VS was relatively unaffected by adaptation, but significantly increased by CD. (iv) Gain exhibited large decreases due to adaptation but less so in response to CD. The relative decrease in gain was particularly small at 20 Hz. All effects in B and C are means and standard errors. 5 Hz: $n = 13$ afferent neurons from $N = 12$ individuals; 20 Hz: $n = 22$, $N = 21$; 40 Hz: $n = 15$, $N = 14$. The data and code underlying this figure may be found at DOI: 10.6084/m9.figshare. 13034012. CD, corollary discharge; VS, vector strength.

we observed the same effect during the spontaneous and evoked periods indicates that the CD protected the vesicle pool from evoked release (Fig 5A). The phase interval exhibited complex effects during the swim bout. Early spikes were initially delayed but then steadily advanced (0.1 $q$, $p = 0.02$), whereas late spikes became steadily delayed to a peak in the post-swim period (0.9 $q$, $p = 0.008$). Overall, there was no statistically discriminable effect on the median phase (0.5 $q$, $p = 0.31$; Fig 5B). The changes in response phase resulted in an initially higher vector strength that decreased to a minimum after the cessation of swimming ($p = 0.005$; Fig 5C). The lower vector strength after the swim is consistent with preserving a wider sensory band width at rest. Altogether, the changes resulted in maximum stimulus gain in the post-swim period ($p < 0.001$; Fig 5D). Together, these results point to a preservation of responses to post-swim stimuli and that CD protected lateral line sensors from adaptation-induced losses in sensitivity.

### Heterogeneity in adaptation and responses to corollary discharge

We found considerable heterogeneity across all units at all frequencies, which prompted us to examine how heterogeneity among sensors resulted in differing responses to CD inhibition. We hypothesized that heterogeneity in adaptation rates [10] results in more complex responses to CD than revealed by average responses alone. We used cluster analysis to categorize types of lateral line responses according to the observed effects of CD on spontaneous and evoked spike rates ($R_S$ and $R_E$, respectively, depicted in Fig 3). The use of $R_S$ provided an independent measurement of inhibition when responses were not influenced by adaptation (as outlined in Fig 1Bii). We found 3 distinct response types (Fig 6Ai). The first 2 groups exhibited consistent levels of inhibition between spontaneous and evoked spike rates ($R_S = R_E$, Fig 6Aii). In the third group, although spontaneous spike rates were strongly suppressed (low $R_S$), we observed only weak inhibition of evoked spike rates (high $R_E$; Fig 6Aii). A linear regression model indicated that variation in $R_E$ could be explained just by the interaction between $R_S$ and the level of adaptation ($R_S$: $p = 0.20$, CI: −0.08 to 0.36; adaptation = Last/First response: $p = 0.31$, CI: −0.06 to 0.19; $R_S \times$ adaptation: $p < 0.001$, CI: 0.33 to 0.94). Adaptation was weakest in group 1 and strongest in group 3, which respectively corresponded to high and low $R_E$ despite the same $R_S$ (Fig 6Aiii). Adaptation was intermediate in group 2. We infer that group 2 responses are composed of a mixture of weakly and strongly adapting responses, which is supported by the importance of the interaction term in the linear model.

The afferent response types differed in their magnitude and phase characteristics of spiking. Groups 1 and 2 exhibited lower spike counts per stimulus compared with group 3. As followed from the clustering procedure based on the activity ratios, CD ON greatly depressed spike responses only in group 1, moderately in group 3, and very little in group 2 (Fig 6Bi). With CD OFF, groups 1 and 2 exhibited wider response intervals compared with group 3 (Fig 6Bii). With CD ON, all response intervals were narrowed, although to different extents. Additionally, the phases of early (0.1 $q$) and median spikes in groups 1 and 3 were delayed and substantially so in group 1 (Fig 6Bii). This resulted in increased vector strength in all groups, particularly group 1 (Fig 6Biii). Altogether, CD substantially reduced the gain of group 1, reduced group 3

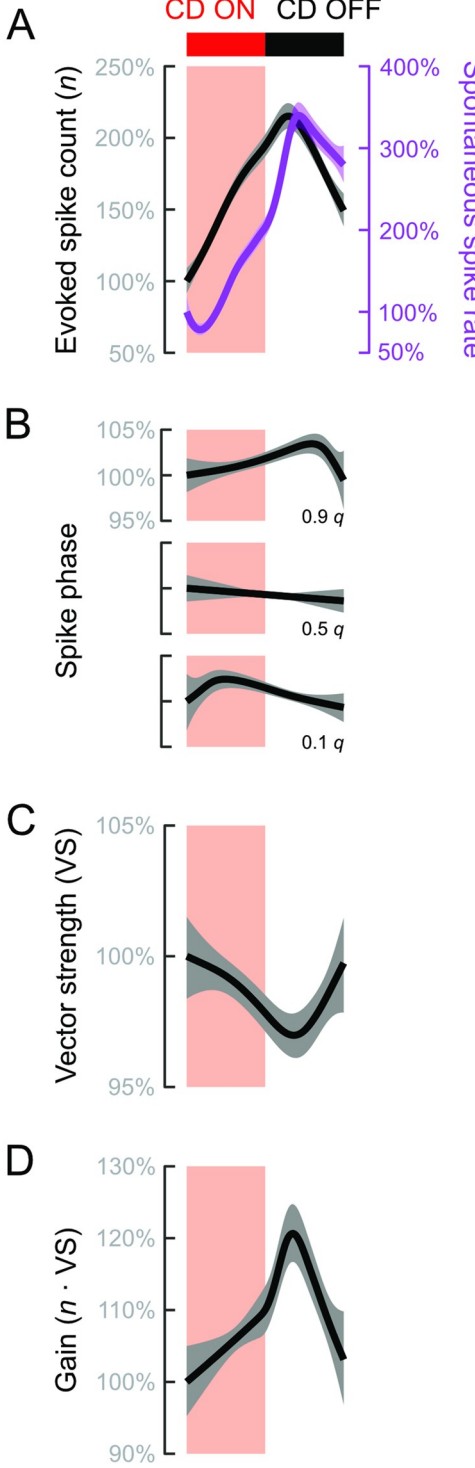

**Fig 5. Time course of evoked responses within and immediately after swim bouts.** All response were normalized to the start of the swim. **(A)** Evoked spike counts and spontaneous spike rates (green) increased to a peak after the end of the swim. **(B)** Response phases exhibited a complex pattern over the course of the swim bout. Early (0.1 $q$) and late (0.9 $q$) spikes exhibited opposite trends. Median spike phase (0.5 $q$) remained nearly constant over the swim duration. **(C)** VS declined to a minimum in the post-swim period, which is related to the opposite trends of the 0.1 and 0.9 $q$. **(D)** Gain increased over the swim to a maximum in the post-swim period. $p < 0.001$ for all smooth functions in each panel. The data and code underlying this figure may be found at DOI: 10.6084/m9.figshare.13034012. CD, corollary discharge; VS, vector strength.

to approximately unity gain and did not change, or even slightly increased, the gain of group 2 (Fig 6Biv). Thus, the heterogeneity among cells leads to classifiable differences in responses to stimulation and CD ON.

## Corollary discharge homogenizes and linearizes lateral line sensor feedback

We developed a simplified computational model of lateral line sensors to probe how cellular heterogeneity in adaptation rates and strengths of CD gives rise to variation in spike responses. In our 2-state Markov model (Fig 7A), a sensor may be currently sensitive or insensitive

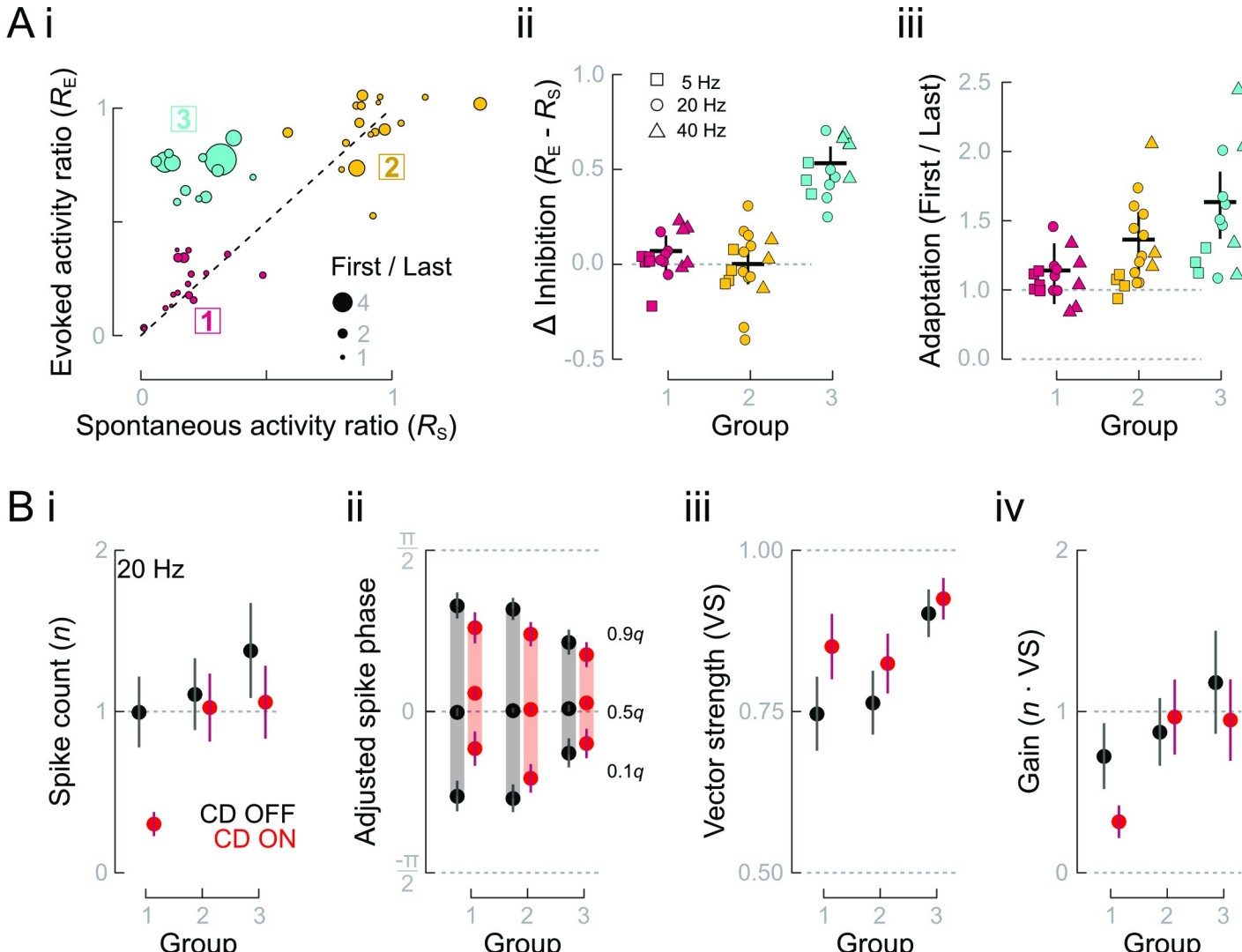

**Fig 6. Heterogeneous adaptation rates and CD strengths result in distinct afferent neuron response types. (A)** (i) The ratios (CD ON/CD OFF) of evoked and spontaneous activities ($R_E$ and $R_S$) revealed 3 distinct clusters of response types. (ii) $R_E$ and $R_S$ were equal in groups 1 and 2 ($\Delta$ Inhibition = 0) but diverged in group 3. (iii) The extent of adaptation increased from group 1 to group 3, shown as First/Last responses to highlight the range of adaptation. **(B)** Evoked activities and responses to CD differ among response types at 20 Hz. (i) Only groups 1 and 3 exhibited significant CD inhibition, as followed from the cluster analysis. (ii) Group 1 and 2 responses had wider response intervals than group 3, but all groups' response intervals were narrowed by CD. Additionally, CD ON resulted in a delayed median phase of groups 1 and 3, compared with the median phase of CD OFF. (iii) The larger response interval widths of groups 1 and 2 resulted in lower vector strength that was more impacted by CD, compared with group 3. (iv) Gain was substantially reduced in group 1 responses reduced and constant or slightly increased in groups 2 and 3, although still approximately unity. Data and code underlying this figure and details of statistics and group comparisons may be found at DOI: 10.6084/m9.figshare. 13034012. CD, corollary discharge; VS, vector strength.

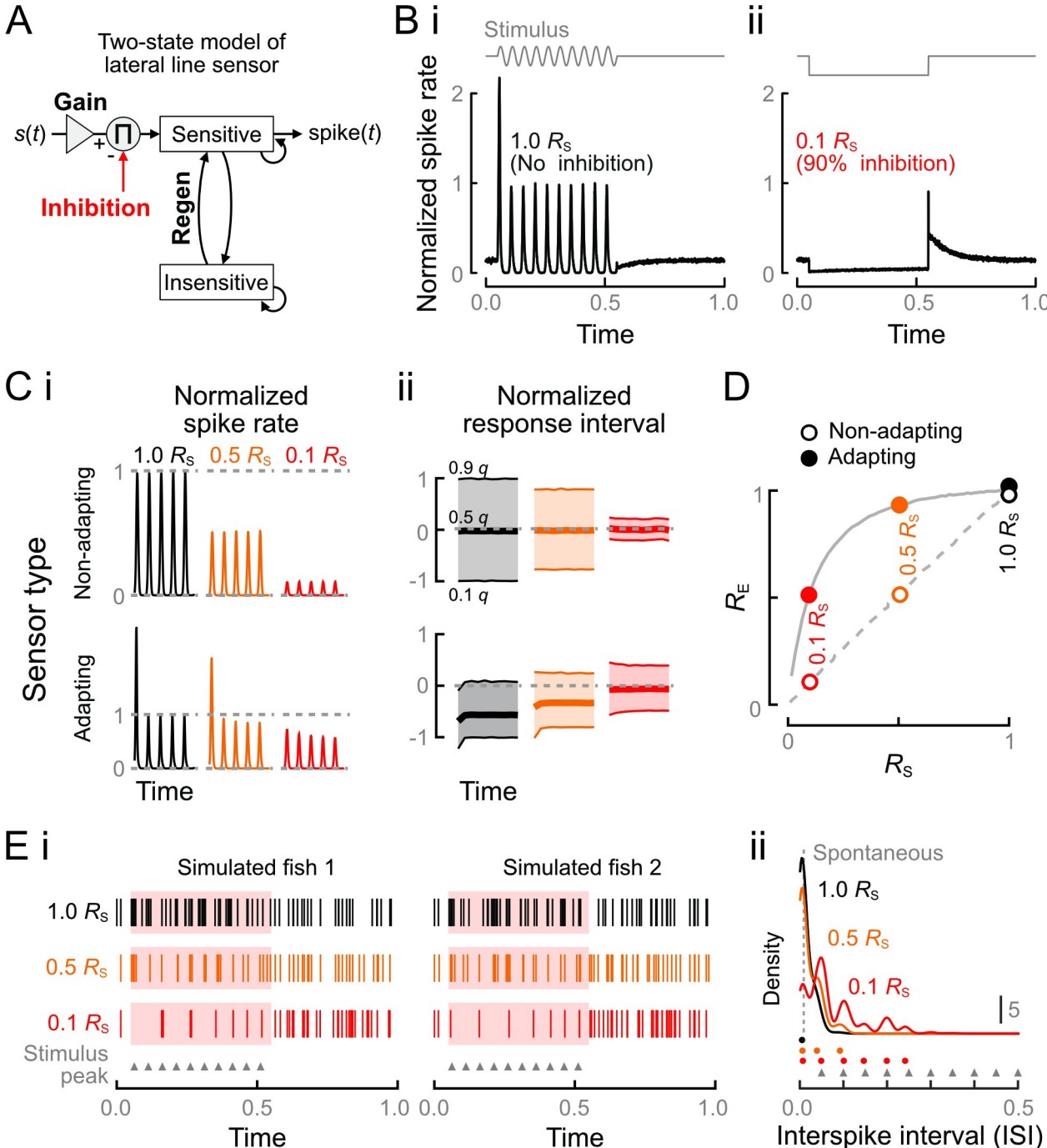

**Fig 7. Computational model of the effects of heterogeneity on lateral line sensing. (A)** (i) Schematic of the 2-state model of lateral line feedback. The input stimulus is transduced by a sensor with variable gain and regeneration rates. The simulated sensor responds probabilistically to the stimulus only while in its sensitive state, after which it transitions to and remains in the insensitive state until regeneration. Tuning of the gain and regeneration constants was used to model adapting and nonadapting response types when subject to inhibition of strength $R_S$. **(B)** Simulated responses of an adapting sensor to stimulation with no inhibition (i) or to strong inhibition in the absence of stimulation (ii). Our model recapitulated the observed changes in spike rate in each condition. After stimulation, the sensor exhibited a refractory period before recovering spontaneous spike rates, whereas after suppression it exhibited rebound spiking. **(C)** Adaptation rates determine the dependence of spike count (i) and phase interval (ii) on inhibition. Nonadapting responses were constant over time and both spike count and response intervals were reduced proportionately to $R_S$. Responses of adapting sensors were nonlinear over time and with respect to inhibition strength, although the nonlinearities were largely suppressed by the strongest level of inhibition (0.9 $R_S$). Spike counts and phases were normalized to the steady states with 1.0 $R_S$. **(D)** Nonadapting responses maintained $R_E = R_S$ for all $R_S$, whereas $R_E > R_S$ for all $R_S < 1$ in adapting responses, supporting the observed response clusters in Fig 6. Dashed and solid gray lines denote the trajectories of nonadapting and adapting responses at inhibition strength intervals of 0.02.

Values at $R_S = 1$ are staggered for visibility. **(E)** Simulated feedback from a population of heterogeneous sensors. (i) In the absence of inhibition (1.0 $R_S$), stimulus phase was difficult to discriminate. Strong inhibition (0.1 $R_S$) resulted in regular feedback closely corresponding to the stimulus peaks (triangles). (ii) The stimulus frequency was poorly reflected in the ISIs at 1.0 $R_S$, but at 0.1 $R_S$, we observed prominent peaks corresponding to integer multiples of the stimulus frequency. The ISIs corresponding to density peaks at each $R_S$ are denoted by circles. Vertical dashed line is the ISI of spontaneous spiking activity. The data and code underlying this figure may be found at DOI: 10.6084/m9.figshare.13034012. ISI, interspike interval.

(conceptually similar model to [72]). The probability that a stimulus $s(t)$ will evoke a spike within a given time interval is conditional on the sensor being in the sensitive state and is given by

$$\log P_{\text{spike}}(t_{\text{spike}} \in [t, t + \Delta t]|\text{state} = \text{sensitive}) = K_1 \cdot s(t) + K_2 \cdot s_0, \qquad (1)$$

where $K_1$ and $K_2$ are constants determining the gain for the stimulus and the strength of the CD, respectively. The CD strength $K_2$ directly multiplies the spontaneous spike rate, $s_0$, which results in a decrease of baseline activity in the absence of an applied stimulus, consistent with the previously observed inhibition of spontaneous spiking [48]. Nonlinearities in the system may result in unequal $K_2$ and $R_S$, so we used the observed $R_S$ to calibrate our results. The expression of log spike probability is a constraint ($P_{\text{spike}} \in (0, 1]$) to enforce a finite spike probability. The exception is immediately after a previous spike, in which case the system is insensitive, and we set $P_{\text{spike}} = 0$ until the system probabilistically regenerates to sensitivity. Together, variation in $K_1$, $s_0$, and the regeneration rate give rise to the system's nonlinearities. Adapting responses were observed when the gain greatly exceeded regeneration rates, which could be interpreted as vesicle pool depletion. Conversely, nonadapting responses occurred when the regeneration rate greatly exceeded the gain, interpretable as the capacity to draw on a rapidly replenishing vesicle pool. Low $s_0$ increased the magnitude of observed adaptation, which could be interpreted as a leak conductance that determines the probability of the system being sensitive at the start of stimulation. Nonetheless, like a similar previous Markov model [72], the construction is general and does not require specific assumptions about hair cells, afferent neurons, or the cause of adaptation.

Simulated lateral line responses captured the principal dynamics observed in this and prior work on hair cells and afferent neurons. The adapting sensor responded strongly to the first stimulus interval of the input but weakly thereafter (Fig 7B; compare peristimulus time histogram (PSTHs) of experimental and simulated responses in S2 Fig). After stimulation, the sensor exhibited a refractory period before regaining steady spontaneous spike rates. Without evoked spiking, inhibition caused depressed spontaneous spike rates that increased over time (compare to Fig 5A) and the release of inhibition was followed by rebound spiking (Fig 7B). Rebound spiking has been observed when hair cells are hyperpolarized by natural or galvanic activation of efferent fibers (Fig 5A) [39,48] or after hair cells are released from hyperpolarization caused by sustained antisense deflection of neuromast cupulae [10]. In the case of cholinergic inhibition, the magnitude of this rebound may be somewhat blunted by the slower time constant of acetylcholine reuptake [48] compared with the instantaneous cessation of inhibition in our simplified model.

The simulations allowed us to precisely examine the magnitude and phase of sensor responses with respect to the stimulus interval in a way that was not possible in our experiments. The feedback from nonadapting sensors was of constant magnitude and phase throughout the stimulus period (Fig 7C; responses truncated to first 5 stimulus intervals). The nonadapting sensor responses were centered on $\pi/2$ but with a wide response interval, which resembled the predicted idealized response (Fig 2C). Strong inhibition greatly narrowed the response interval (equivalently, increased vector strength). Despite the reduction in the magnitude of responses, so that even single spikes from the nonadapting sensors will faithfully

represent the stimulus. Adapting responses exhibited the experimentally identified phase shift as well as a large phase advance relative to peak stimulation as predicted in Fig 2. However, strong inhibition (0.1 $R_S$) largely suppressed the nonlinearities of both the spike count and phase intervals. Importantly, strong inhibition delayed adapting sensors' median phase to coincide with the peak of stimulation at $\pi/2$. We conclude that strong CD homogenizes and linearizes the outputs of sensors with highly heterogeneous adaptation rates.

We next examined the interactions between levels of adaptation and the effects of CD on spontaneous and evoked spike rates. We found that for nonadapting responses, $R_E/R_S = 1$ for all $R_S$, whereas for the adapting responses, $R_E/R_S > 1$ for all $R_S < 1$ (Fig 7D). Notably, $R_E/R_S$ converged at high $R_S$, i.e., with low levels of inhibition, supporting our assertion that group 2 responses represented weak inhibition of a mixture of adaptation levels (compare to Fig 6). At 0.1 $R_S$, equal to a 90% inhibition of spontaneous activity, the adapting sensors maintained more than 50% of evoked activity. This supports our distinction between groups 1 and 3 as well as the magnitude of adaptation as an underlying cause.

Our central hypothesis was that CD enables proprioception by restructuring reafference into predictable feedback. We finally addressed this hypothesis by simulating feedback from a population of heterogeneous lateral line sensors with varying levels of inhibition. Afferent feedback from 2 representative, simulated "fish" is shown in Fig 7Ei with inhibition equal to 1.0, 0.5, and 0.1 $R_S$. In the absence of inhibition (1.0 $R_S$), the mixtures of responses from heterogeneous sensors resulted in elevated spike rates but with poorly distinguished phase, even between the sense and antisense portions of the stimulus cycle. The evoked interspike intervals (ISIs) were even shorter than those of spontaneous spiking and therefore dominated by periods less than 1 stimulus cycle (Fig 7Eii). Strong CD (0.1 $R_S$) resulted in a sparsely coded stimulus (Fig 7Ei). The density at the first peak, corresponding to repeated spikes within cycles, was greatly reduced (Fig 7Eii), and, in fact, responses to many peaks were suppressed altogether. The remaining spikes occurred close to the reafferent stimulus peak at $\pi/2$, resulting in ISIs that were integer multiples of the stimulus frequency. In contrast, with 0.5 $R_S$, we observed only a minor increase in the frequency of ISIs corresponding to the stimulus frequency. The small change is attributable to generally marginal effects of 0.5 $R_S$ on the magnitude and phase interval of evoked responses (Fig 7C and 7D). We speculate that these limited effects explain the rarity of intermediate levels of inhibition strength within the lateral line (Fig 6A; first noted by [49]).

## Discussion

We provide a multiscale neuromechanical model that elucidates how CD can mediate the proprioceptive information content of lateral line feedback during axial undulation. Axial undulation is the ancestral mode of vertebrate locomotion, predating paired appendages, and is defined by a traveling body wave that enables propulsion through complex environments. To stabilize this mode while navigating, undulating fishes must possess a sense of self-motion and body position, yet they lack many of the classes and much of the distribution of proprioceptors found in other vertebrates [19–22]. Early studies recognized the potential for proprioception from the lateral line but could not provide conclusive evidence (reviewed in [12,13]). Proprioceptive functions would be difficult to detect without a hypothesis for how kinematically induced flows can be transduced and related to changes in motor control (Fig 1B). In our neuromechanical model (Fig 2), deflection of the neuromast cupula is coupled to the progression of the mechanical body wave. Because the lateral line can encode stimuli with millisecond precision [8], there is the potential for reafference to encode details of the body configuration. However, in the absence of CD, sensors' variable gains and nonlinear responses will reduce the

coherence between afferent spikes and the body wave peak (Figs 2, 4, and 7). Thus, despite periodic spiking in time with body motions [12,13], the wide response interval will reduce the information content and proprioceptive value (see also [8]). We examined how CD within the lateral line microcircuit modulates the feedback that is transmitted to the brain (Fig 1Bii). We find that CD enables peak detection within the peripheral nervous system. The computation occurs through a synergistic interaction with hair cell polarization that narrows response intervals around the peak of deflection (Figs 2 and 7). The resulting feedback faithfully encodes the stimulus peak and frequency despite the biophysical heterogeneity of the sensor population (Fig 7E).

Our experiments and simulations suggest that CD cancels the unpredictable parts of reafference, rather than all reafference. Numerous examples of complete cancellation have been documented in animals, and perhaps for this reason, it has been presumed that inhibitory modulation of the lateral line follows a similar principle [3]. Nonetheless, as Bell [54] noted, simple mechanisms of cancellation become problematic for complex and long-duration motor actions and even more so when the reafference carries vital feedback. An alternative is adaptive cancellation, which compensates long-duration and low-frequency reafference-like respiration [27,54,55] but which cannot compensate the subsecond and high-frequency reafference typical of swimming. A second alternative is to filter the reafference so that only specific features of reafference are transmitted. This occurs in the lateral line and the phylogenetically related vertebrate auditory system, in which reafferent feedback is reduced but not eliminated [48,50,73]. The resulting reafference can be critical for proper motor output [12,28], such as the importance of the auditory system in phonation [56]. However, filtering the reafference may be challenging with heterogeneous sensors that cause the magnitude and temporal properties of feedback to vary over time.

We examined sensor heterogeneity through differences in adaptation rates and varying responses to CD inhibition (Fig 6). We found a wide range in adaptation rates, as is typical of hair cell systems [11], which could be due to numerous sources, including the age and maturity of the hair cells and afferent neurons [25,61,66]. We also found bimodal inhibition of evoked spike rates ($R_E$) as previously documented [49], but our analyses suggest that inferring CD strength from $R_E$ is not straightforward. A given level of inhibition independently measured on spontaneous spike rate ($R_S$), resulted in divergent $R_E$ depending on level of adaptation (groups 1 and 3 in Fig 6A, trajectories in 7 D). Using $R_E$ to determine afferent neuron polarity is currently speculative because we do not know how afferent neuron variability, including in age, size, and input resistance [9,25], transforms hair cell feedback. In addition, the previous analysis of $R_E$ was performed on hair cells at lower stimulus frequencies [49], which will experience much more gradual adaptation [10]. Nonetheless, it is interesting to speculate that if group 1 are anterograde sensors (rostrocaudal, on the basis of their high $R_E$), then a molecular mechanism may not be required to explain selective inhibition [49]. Rather, if the magnitude of efferent polarization is equal in all hair cells (measured by $R_S$) but anterograde sensors are weakly adapting (suggesting generally low gain), then we would expect to see altogether stronger inhibition of the reafference experienced by forward swimming. The lower adaptation and stronger inhibition will also result in narrower response intervals that strongly correspond to the peak of reafferent stimulation (Fig 7). Anterograde flow is particularly well represented by activity throughout the brain [62], so more precise feedback will enable a wider range of computations [7]. On the same basis, groups 2 and 3 may be retrograde (caudorostral) sensors. In this polarity, the mixture of high and low gains and adaptation rates may facilitate rapid initial responses along with constant sustained responses while escaping a suction source, as when attacked [74,75]. Among all groups, we find that CD suppresses the temporal effects of heterogeneity so that the transmitted output is a consistent function of the stimulus (Fig 7).

The capacity of CD to restructure feedback from afferent units with high gain and large amounts of adaptation depended on very strong inhibition (Figs 6 and 7). Why are all hair cells not suppressed equally strongly? Our findings support the hypothesis that CD evolved to suppress afferent spiking as much as necessary, but not more [49]. One consequence of universally strong CD inhibition would be the elimination of weaker reafference signals from retrograde flows that are relatively weaker during forward swims [49]. A second consequence is that strong inhibition would lead to post-swim artifacts. In the absence of stimulation, CD ON caused post-swim rebound spiking (Figs 5 and 7), similarly to naturalistic and galvanic activation of cholinergic fibers in the auditory system [71] or sustained antisense deflection of hair cells [10]. Excessive inhibition that causes rebound spiking in the retrograde direction could be misinterpreted as a predator attack from behind. A requirement of the CD is therefore that it be calibrated to control motion artifacts without introducing new ones. Further factors that contribute to preventing artifacts are the continuous post-swim decline of both stimulation and inhibition after the swim (glide and acetylcholine reuptake [48]), as opposed to the abrupt changes between CD ON and OFF modeled here (Fig 7B). Time-dependent changes in inhibition strength [49,50] may smooth out the intense reafferent signals at the swim start that are associated with stronger first tail beats [51], anterograde flow [49], or the kinetics of the rapid vesicle pool [59].

Even with the CD, stimulation greater than the expected reafference will result in adaptation, and this could be advantageous for exafference sensing. Adaptation is a form of short-term synaptic plasticity that contributes to functions like source localization through bilateral comparisons or between pairs of detectors [11,40,41]. In a related manner, adaptation could contribute to rheotactic behaviors, in which fish orient upstream by comparing bilateral differences in water velocity, presumably encoded by differences in spike rate [29,45,75]. Bilateral differences in flow will cause differential adaptation, and as a logical extension of our results, bilateral differences in spikes' temporal code [8]. Our results suggest that CD's effects on spike rate and phase will improve fishes' sensitivities to bilateral flow differences. A critical next step for understanding the role of reafferent feedback in motor control will be to unravel the nature of the controller in the central nervous system (Fig 1Bi). Studies of freely behaving fish will be essential, likely including optogenetic dissection of the CD's role in swimming feedback control and natural swim motions. Similarly, the actual magnitude and temporal dynamics of cupular deflection during swims is unknown, even though the coherence and encoding capacity of lateral line feedback critically depends on both the frequency and amplitude of stimulation [8,23]. For example, the boundary layer mechanically filters the frequencies and stimulus amplitudes that are passed to the cupula [8,23], which might have contributed to the higher average gain for 20-Hz stimulation than 5 or 40 Hz (Fig 4). Moreover, the amplitude and frequency content of reafference changes over the course of swim bouts (as shown in Fig 2A, [76]), suggesting that changes in inhibition strength during and after swims [35,48–51,77] may contribute to precise and faithful preservation of stimulus phase and signal properties. We have described how feedback from external proprioceptors can allow fish to sense and control the continuously varying body configuration during aquatic locomotion. More generally, our findings point to a sense of body awareness coupled to the mechanical environment that may be widespread among animals inhabiting fluid environments.

## Materials and methods

### Electrophysiological recordings

Adult zebrafish (*Danio rerio*) were obtained from a laboratory-bred population and housed as described previously (e.g., [9,24,48], in accordance with protocol 201903267 approved by the

University of Florida's Institutional Animal Care and Use Committee). Protocols were based on our previous study [48]. Larvae from 4 to 6 dpf were paralyzed by immersion in 1 mg/mL α-bungarotoxin (Reptile World Serpentarium, St Cloud, Florida, United States of America) diluted in 10% Hank's extracellular solution [48] and then washed and replaced in extracellular solution until experiments (>30 minutes). α-bungarotoxin blockade of the neuronal-type nicotinic acetylcholine receptor (nAChR) α9 found in neuromasts is reversible after a brief washout period [78]. Larvae were pinned laterally to a Sylgard-bottomed dish using etched tungsten pins inserted through the dorsal notochord at the anus, at a second location immediately caudal to the yolk sac (approximately at neuromast L1), and through the otic capsule. The preparation was then covered in extracellular solution for recording. Fish health was monitored by blood flow.

We reliably obtained single-unit afferent recordings, or occasionally 2 clearly differentiable units, using borosilicate glass pipettes with diameters approximately 15 μm. At this pipette diameter, it was necessary to approach the ganglion under high pressure (−150 to −200 mm Hg) to break in. Once a unit was located, pressure was released sufficiently to maintain a stable recording (typically between 0 and −75 mm Hg). In most cases, the afferent recording was stable for hours. Afferent recordings were performed at 20 kHz with 50 V/mV gain, Bessel filter of 1 kHz, and an AC high-pass filter of 300 Hz. Signals were amplified through an Axon Instruments CV-7B head stage (Molecular Devices (MD), San Jose, California, USA) and an Axoclamp 700B amplifier (MD), then recorded with a Digidata 1440A digitizer and pClamp10.7 (MD). Spike times were obtained by thresholding in Clampfit 10.7 (MD). Once an afferent unit was identified, we systematically probed neuromasts for the stereotypical afferent evoked response to a periodic stimulus [24] using a glass bead attached to a single-axis piezo stimulator (30V300, Piezosystem Jena, Hopedale, Massachusetts, USA). Dipole fluid motions of this type are commonly used to assess near- and far-field responses to oscillatory stimuli, such as from prey or conspecifics [29,37,74]. The dipole vibrating at 20 or 40 Hz simulates neuromast stimulation by the larva's own movements, because reafference in vivo is associated with alternating [12,13] rather than direct currents [49,50]. Stimulation at 5 Hz was also performed to compare responses to a frequency at the extreme lower limit of typical tail beats [51]. The probe head was roughly spherical with a diameter of 56.6 μm, formed by melting the tip of a recording pipette using a MF-830 microforge (Narishige International, Amityville, New York, USA). Probing started at D1 and continued down the body until the evoked response was elicited. The majority of units could be stimulated by D1 ($n$ = 16) or D2 ($n$ = 3) and were classified as D-type, and only a few units did not exhibit responses until we reached L1 ($n$ = 1), LII.1 or LII.2 (each $n$ = 2), or LII.3 ($n$ = 1), classified as L-type. In one case, we were unable to locate any attached neuromast. Note that, because we only identify the most proximally innervated neuromast, the afferents could have been connected to more distal neuromasts as well [25]. After identifying the neuromast, the probe was placed 1 diameter away from the cupula, and stimulation was performed with a constant approximately 8-μm amplitude. At no point was the cupula itself touched, which substantially alters responses of the hair cell-afferent group [24].

Once an afferent and associated neuromast were identified, we obtained extracellular recordings of the ventral root (VR) arborizations in the myotomal cleft [48]. Pressure in the pipette was manually controlled through a syringe. VR recordings were acquired as per afferent units, but with a gain of 100 V/mv. Swim bouts were preprocessed by thresholding the VR signal in Clampfit 10.7 and then manually curated using custom software in R 3.6.0 [79]. VR recordings were stable for approximately 15 to 60 minutes. All swims were spontaneous or elicited by a light flash.

## Data processing

We recorded 100 sweeps with the stimulus on for 1 second then off for 2 seconds, which ensured recovery of afferent spike rate in all but one possible case at 40 Hz. We first examined responses at 20 Hz ($n$ = 22), then 1 or both of 40 Hz ($n$ = 15) or 5 Hz ($n$ = 13) depending on preparation stability. Additional recordings were performed as required to guarantee sufficient numbers of swims for analysis. Spikes were labeled as outlined in the main text (Fig 3). The post-swim interval was equal in duration to the swim [48] (S3 Fig). We analyzed evoked responses either as a function of stimulus number or as the average over the stimulation period. Within a stimulus interval of period $T$, the phase $\theta$ of a spike occurring at time $t$ was expressed as $\theta = t/T$. The apparent spike phase depends on stimulus frequency (change in phase location visible in Fig 3B, see also [8,24,67]), which is a consequence of conduction delays in the afferent fibers [24]. For each cell at each frequency, we therefore subtract the median phase during CD OFF, allowing us to compare across cells. The phase can then be used to examine changes in the distributional properties of spikes, such as location and width. The commonly reported vector strength (VS, e.g., (24)) for a set of $n$ spikes is calculated by

$$\text{VS} = \frac{1}{n}\sqrt{\left(\sum cos\theta\right)^2 + \left(\sum sin\theta\right)^2} \tag{2}$$

and is related to the circular variance according to $\sigma^2 = -2log\,(\text{VS})$ [64]. Vector strength is constrained to the interval (0,1], with VS→1 as the afferent response interval narrows. Single spikes' information content is quantified by their gain, calculated by the product of spike count and VS. We additionally computed lower, median, and upper quantiles (0.1, 0.5, and 0.9 quantiles, $q$) of the phases, which reflect the tails of the spike phase distribution while being more robust than the minima (i.e., first spike latency) and maxima.

## Statistical analysis

Differences due to adaptation and swimming were characterized through generalized linear models (GLM). Modeling was performed with the R function *glmer()*, with Tukey post hoc tests performed with the function *glht()* in package *multcomp* [80]. These contrasts were chosen to examine average differences in each condition and to compare our results to literature values (S1 Fig). Significant differences in means were assessed by confidence intervals that excluded 0. All models were fit by maximum likelihood. Arcsin-transformed vector strength was modeled with a Gaussian error distribution, and spike counts and gain were modeled with a quasi-Poisson distribution. The lower phase quantile (0.1 $q$) was cube transformed prior to analysis to stabilize the variance. Throughout, we provide relevant estimate effects and 95% confidence intervals; full statistical models (R summaries), including relevant $p$-values, are included with model code at DOI 10.6084/m9.figshare.13034012. Statistical significance was tested at $\alpha$ = 0.05.

The relationships between $R_E$, $R_S$, and adaptation were explored through linear models and cluster analysis. The level of adaptation was estimated as the difference in responses to the first and last stimulus. We found that directly averaging first and last responses was prone to high variance, likely due to sampling error. For the purposes of classification, we determined the average responses to the first 2 stimulus intervals with respect to the last 2 intervals. The ratios of activities reported in Fig 4 (Last/First and CD ON/CD OFF) were estimated from the predicted responses at each stimulus for each individual at each frequency from the statistical models (modeling described below). The prediction of $R_E$ as a function of $R_S$ and level of adaptation was then studied in a linear mixed effects model (R package *nlme*, [81]), with individual and frequency as crossed random effects. The regression slope was forced through 0 so that the approximate confidence intervals of the predictors could be compared with 0 to assess

significance. Responses of each individual at each frequency were assigned to the 3 clear clusters of $R_S$ and $R_E$ through *k*-means analysis. Post hoc analyses based on this clustering were used to illustrate the magnitudes of differences between clusters and determine average differences in adaptation level.

We used generalized additive models (GAMs) to model the parametric (response class and swim state) and nonparametric (adaptation) factors that contributed to variation in cell responses over time. The parametric portion of the fit was determined by the differences in responses between spikes labeled as CD OFF or CD ON, averaged over the entire interval (inactivity was the reference level), for each response class. The interaction of swim state and class indicated the differential effects of CD ON for each response class (Fig 6B). The nonparametric portion of the fit accounted for the continuous changes over time due to adaptation, which we refer to as the smooth function (shown in Fig 4A [82,83]). The statistical significance of the smooth function indicates whether a nonlinear fit contributes to explaining patterns in the data. We employed an adaptive spline basis set [83] with the upper limit of spline knots set to the frequency (with a 1-second stimulus, the maximum number of knots was the number of response intervals). We allowed for variation in the shape of the smooth function that could influence our estimated parametric effects, either due to response class (factor smooths [83]) or to individual (random effect). We used our models to compare responses across frequencies and afferent response types (groups in Fig 6). The means and standard errors of all quantities across frequencies and response classes were obtained by simulating from the model posterior distributions. We performed 1,000 draws of the first and last stimulus periods and of the average CD OFF and ON responses. We calculated the predicted average effects of adaptation (Last/First response) and CD (CD ON/CD OFF) and compared the probability that distributions differed at $\alpha = 0.05$. We present the results as either the main effects of CD ON and CD OFF (Fig 4) or by response class (Fig 6).

The GAM modeling is also appropriate for examining changes in evoked responses during and after the swim. These 2 intervals are of principal interest in studying the consequences of movement on sensing. Modulation of the pre-swim period was not revealed in our previous analysis of spontaneous spike rates in the pre-swim period [48]. Within each individual at each frequency, we selected swims that fell entirely within (1,650 swims) or outside (3,348) the stimulus interval, which were then aligned by their terminus and normalized to unit duration. We then estimated the time course by averaging over all swim bouts, which will ultimately depend on the amount of data used to estimate the average. Estimates obtained from a larger number of observations were therefore given more weight. We analyzed responses from the swim start up to one swim duration after the swim end as a function of relative time, using an adaptive spline basis. Differences among stimulus frequencies were included as a random effect and a random intercept term was included for individual cells. Spontaneous activity was averaged over time (spike rate) rather than per stimulus period. All statistical summaries and significance values of post hoc comparisons are contained with model code at DOI 10.6084/m9.figshare.13034012.

## Two-state model of lateral line sensing

We explored potential causal mechanisms of the phase relationships discovered experimentally by simulating different biophysical parameter sets giving rise to adapting and nonadapting sensory responses. We modeled the responses of a sensor that transitions between sensitivity (state 1) and insensitivity (state 2). The discrete time transition matrix is given by

$$
\begin{bmatrix} 1 \to 1 & 1 \to 2 \\ 2 \to 1 & 2 \to 2 \end{bmatrix} = \begin{bmatrix} P_A & P_B \\ P_C & P_D \end{bmatrix} = \begin{bmatrix} 1 - P_B & (K_1 \cdot s(t) + K_2 \cdot s_0)\lambda^* \\ K_3 \lambda^* & 1 - P_C \end{bmatrix}.
$$

The transition matrix elements are normalized to stimulus wavelength divided by sampling frequency ($\lambda^*$, simulated at 2 kHz). Spontaneous spike rate, $s_0$, is the number of expected spikes per stimulus wavelength, multiplied by the inhibition constant, $K_2$. The stimulus is modeled by a time-dependent stimulus function, $s(t)$, multiplied by the gain, $K_1$. When $s(t)$ is constant (a time-invariant stimulus), a rapid burst of evoked spikes rapidly settle into a constant rate that reflects the balance of regeneration and release probabilities. For time-varying stimuli, the responses depended on the relative coefficient magnitudes and extent of adaptation. The expected regeneration rate per $\lambda^*$ is given by $K_3$. The parameters sets were chosen to simulate nonadapting and adapting responses (nonadapting: $K_1 = 4$, $K_3 = 0.04$, $s_0 = 40$; $K_1 = 0.04$, $K_3 = 4$, $s_0 = 4$). The stimulus duration was 20 $\lambda^*$, with $K_2 = 0$ for half of each simulated sweep so that the system would re-equilibrate. Inhibition was applied with or without a stimulus, and either $K_2 = 1$, 0.5, or 0.1 (0, 50, or 90% inhibition) or $K_2$ from 0.02 to 1 in increments of 0.02. Simulations were run for 2 to 10k iterations as required to depict smooth responses in Fig 7. The first 2 iterations were discarded as burn-in to avoid any impact of the initialization conditions. We finally simulated the responses of a model fish with a population of 30 heterogeneous lateral line sensors. Variable adaptation rates were generated by uniform sampling of $K_1$ and $K_3$ with constant $s_0$. Each sensor was simulated for a single swim bout and the spike times summed. We calculated ISIs by the time difference between sequential spikes of 1,000 simulated fish.

## Supporting information

**S1 Fig. Parameters of recorded swim bouts. (A)** Histogram of swim bout durations. **(B)** Average spike rate in swim (CD ON), post-swim, and post-stimulus intervals, relative to inactivity (CD OFF). This estimate of the relative evoked spike rate while swimming was obtained by averaging over all swim bouts and individuals, which resulted in a higher estimate than when accounting for differences due to cell class (Figs 4 and 6). The data and code underlying this figure may be found at DOI: 10.6084/m9.figshare.13034012. CD, corollary discharge. (TIFF)

**S2 Fig. Representative PSTH of evoked responses. (A)** PSTH at 20 Hz corresponding to individual and traces in Fig 3. Red bars denote a spontaneous swim bout. Spike rate calculated in 50-ms intervals for spikes labeled as CD OFF. Stimulus trace shown below. **(B)** PSTH of simulated responses. Inhibition was only applied over the entire stimulus interval. Note that responses were normalized to the wavelength period. The data and code underlying this figure may be found at DOI: 10.6084/m9.figshare.13034012. CD, corollary discharge; PSTH, peristimulus time histogram. (TIFF)

**S3 Fig. Representative traces and PSTHs of evoked responses at 5 and 40 Hz.** Black diamonds over afferent ganglion traces demark highly evoked responses in the immediate post-swim period (red hatched intervals, equal in duration to the swim bout). In many cases, evoked activity after a swim was equal to the evoked activity at the start of the stimulus presentation. Spike rates were calculated in 50-ms intervals. Same individual as Fig 3. The data and code underlying this figure may be found at DOI: 10.6084/m9.figshare.13034012. PSTH, peristimulus time histogram. (TIFF)

## Acknowledgments

We are extremely grateful to Dr. Noah J. Cowan for numerous insightful discussions and Drs. Yuriy Bobkov, Paula Duarte Guterman, Miriam H. Richards, and James A. Strother, whose critical evaluation greatly improved the manuscript.

## Author Contributions

**Conceptualization:** Dimitri A. Skandalis, Elias T. Lunsford, James C. Liao.

**Data curation:** Dimitri A. Skandalis.

**Formal analysis:** Dimitri A. Skandalis.

**Funding acquisition:** James C. Liao.

**Investigation:** Dimitri A. Skandalis.

**Methodology:** Dimitri A. Skandalis, James C. Liao.

**Project administration:** James C. Liao.

**Resources:** James C. Liao.

**Software:** Dimitri A. Skandalis.

**Supervision:** James C. Liao.

**Validation:** Dimitri A. Skandalis.

**Visualization:** Dimitri A. Skandalis.

**Writing – original draft:** Dimitri A. Skandalis.

**Writing – review & editing:** Dimitri A. Skandalis, Elias T. Lunsford, James C. Liao.

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
