## [Editor Report · Decision Letter 0]

4 Mar 2021

Dear Dr Skandalis, 

Thank you for submitting your manuscript entitled "Corollary discharge prevents signal distortion and enhances sensing during locomotion" for consideration as a Research Article by PLOS Biology.

Your manuscript has now been evaluated by the PLOS Biology editorial staff, as well as by an academic editor with relevant expertise, and I'm writing to let you know that we would like to send your submission out for external peer review.

Please re-submit your manuscript within two working days, i.e. by Mar 08 2021 11:59PM.

Kind regards,

Roli Roberts

Senior Editor

PLOS Biology

---

## [Decision Letter · Decision Letter 1]

26 Apr 2021

Dear Dr Skandalis,

Thank you very much for submitting your manuscript "Corollary discharge prevents signal distortion and enhances sensing during locomotion" for consideration as a Research Article at PLOS Biology. Your manuscript has been evaluated by the PLOS Biology editors, an Academic Editor with relevant expertise, and by four independent reviewers.

You'll see that all of the reviewers think it’s an important question, but three out of the four raise concerns about the lack of clarity and the need for improved presentation; this is particularly important, given the broad readership of our journal, so please do invest significant effort in addressing these concerns. Reviewer #3's assessment is somewhat more negative, but given the positivity elsewhere, we are happy to consider your manuscript further.

In light of the reviews (below), we are pleased to offer you the opportunity to address the comments from the reviewers in a revised version that we anticipate should not take you very long. We will then assess your revised manuscript and your response to the reviewers' comments and we may consult the reviewers again.

We expect to receive your revised manuscript within 1 month.

**IMPORTANT - SUBMITTING YOUR REVISION**

*Resubmission Checklist*

*Published Peer Review*

*PLOS Data Policy*

*Blot and Gel Data Policy*

Sincerely,

Roli Roberts

Senior Editor,

rroberts@plos.org,

PLOS Biology

REVIEWERS' COMMENTS:

Reviewer #1:

[identifies himself as Leon Lagnado]

Skandalis et al. 

Corollary discharge prevents signal distortion and enhances sensing during locomotion

This paper addresses a general and important issue in sensory neuroscience: how does the nervous system deal with stimulation of sensory systems resulting from the animals own motor behaviour rather than events in the external world. This issue has been investigated in a number of contexts, such as the fly visual system, but here the authors investigate mechanical sensing in the lateral line of fish, where it has long been known that motor activity is associated with a copy of the motor signal (a "corollary discharge") delivered directly to the primary receptors (hair cells) through efferent fibers in the sensory organ (neuromast) that senses water flow over the body (effectively a form of proprioception). It has long been known that this CD inhibits hair cell activity and recently it was demonstrated that this inhibition is to a large degree selective for the hair cells that are polarized to be most strongly activated during forward swimming motion. This work now links this information into a bigger picture by developing "a data-driven model integrating swimming biomechanics, hair cell physiology, and corollary discharge to understand how sensory modulation is calibrated during locomotion in larval zebrafish". The authors use this model to test the hypothesis that a major effect of the CD is improve the temporal accuracy with which the amplitude and speed of undulatory body motions are signalled by the lateral line. This hypothesis is based on the idea that the CD will counteracting the phase distortions caused by the heterogenous rates of depressing adaptation in different hair cells within the neuromast and narrow the time-window over which afferent fibers are active by increasing the threshold for activation of synaptic transmission from hair cells.

This is an excellent paper which makes a decisive contribution and deserves a wide audience. It is conceptually rich and does an excellent job of integrating hair cell biophysics into the bigger picture of the lateral line in action. The central hypothesis is interesting and reasonable and the experiments designed to test it are unusually direct, demonstrating that CD does indeed improve timing accuracy in at least some afferent fibers. The discussion of the results is nuanced and thoughtful and again emphasizes the wider functional role of the lateral line system. 

Major criticisms: None. The experiments are designed well and analysed appropriately. The interpretations of the results are reasonable. Wider discussion of results are also balanced and reasonable.

Minor comments

Line 168: Fig. 1Ai and ii should be Fig. 2Ai and ii I think

Fig. 3C: what are the values of the scale bars? Are these averages over time? Please make legend clearer, I had trouble with this.

Reviewer #2:

In this manuscript, Skandalis and colleagues present a new analysis of the role of corollary discharge in mitigating self-generated sensory stimuli (reafference). Their central idea is that corollary discharge effectively narrows the temporal window for sensory inputs so that (a) excessive responses are diminished, and (b) the responses that do occur are tightly timed with respect to the body bend, permitting functional proprioception.

This idea is very interesting and speaks to a vital topic in the field of motor control, and their data and analyses seem compelling. The concept of the role of motor feedback being to tighten sensory responses and limit habituation is quite powerful. At the same time, the presentation of the data was very difficult to follow, and the final model in particular was baffling as it seemed to combine both hair cells and afferents into a single element. If the authors are able to address these concerns, this manuscript would be of interest to a broad audience.

Major issues

1. There are many comparisons made between various aspects of the data, and they are presented at too extracted a level (initially) for readability. Evoked vs spontaneous, early vs late in a train, CD or no CD, four different groups, and frequency of stimulation all make their first appearance in Fig 2. It's hard to know what to focus on. This reader, at any rate, would benefit if these were introduced more slowly, perhaps with an early figure exclusively covering habituation during CD off periods, and a later figure showing the effect of swim and/or frequency, or some other order. The subtraction of delta S and delta E (2Bii) seemed unnecessary, since those data are already compared in 2Bi, and the use of delta S, delta E, delta % (for habituation), and delta CD was taxing to keep in mind.

2. A key claim in the results is that the effect of CD is to narrow the time window for evoked stimulus responses. In the examples presented in Fig 3A, however, there are a lot fewer responses for CD on than for CD off (red dots compared to gray). I was skeptical that the narrower distribution seen in some of these plots actually reflected a narrower response range, because fewer points will undersample the range. Can the authors present a statistical comparison of the variance during the two conditions?

3. The authors separate the recorded afferents into four groups. Group IV only has two cells, both of which were recorded at 5 Hz, which seems to make for a less compelling group identity. How important are these groups? The authors do not discuss whether any of these groups correspond to afferents with the opposite directional tuning, which are not subject to the same suppression during swim (Pichler and Lagnado 2020). The authors state that they could not determine directionality of responses in the Methods, though I was unclear why. The effects of CD on all four groups aren't ever broken out, as best as I can tell, in Fig 3, which raises the question of why divide into those groups in the first place. 

4. The model presented in Figure 5 appears to conflate hair cells and afferents. Throughout the manuscript, the authors have been referring to vesicle release from hair cells, so that was what I thought was being modeled here—but then they speak of spiking, whereas hair cells are non-spiking. Thus it seems this is a model of afferent release of vesicles, but I don't understand how that can be connected with the presented data, since the authors aren't recording from neurons downstream of afferents. Presumably I'm missing something here.

5. There were a number of confusing or incorrect elements in the figures. (a) There seemed to be a mismatch in the data presented in Fig 2Bi and 2Biii. There is a large orange circle in 2Bi, representing >2 spikes difference in early vs late spiking, but there is no equivalent height dot in Fig 2Biii (it should be up near the height of 2x the scale bar, if I understand correctly.) (b) What are the red lines in Fig. 2Aii? In supp Fig 1 these appear to represent periods of swim, based on the red "swim" label, but this graph is explicitly labeled CD OFF so that can't be correct here. (c) Is there any meaning to the vertical spacing of dots in Fig 3A? They seem oddly distributed, e.g. sometimes bimodal, and I couldn't tell if there was a reason for this. (d) What are the two rows of data in Fig 3Div? First and last spikes? It's not described in the legend. (e) "Sample sizes as in Fig 1" (Fig 3, legend) presumably should be "Fig 2"?

Minor points

1. The abstract sounds as though this manuscript is only a model, and doesn't contain new data. Please revise to alert readers to the experimental setup and results.

2. It was a bit difficult to understand why early spikes would be delayed and late spikes advanced by CD, line 135. Can the authors clarify if this is an iceberg effect or something else?

3. In Fig. 2Aiii, various periods are labeled as delta S. However, I thought delta S referred to a subtraction of the spont data during swim minus the spont data outside of swim? This could be more clear.

4. Line 191, does "over time" mean from early to late in the stimulus train?

5. Line 219-220, "Habituation had a small effect on spike count…" doesn't appear to be what's quantified in Fig 2B, and there is no Fig 2C. Perhaps Fig 3 was intended?

6. Line 241, should CD OFF be CD ON?

7. At the end of the swim bout, fish are still gliding forward. Can the authors comment on what this means for their post-swim data? Is it irrelevant because the forward motion is insignificant for the lateral line sensors?

8. Line 269, what is "significance of smooth" both here and elsewhere? What statistic is this?

9. The presentation of data in Fig 3Div was very challenging. Are these all percents? Delta (%) suggests yes, but then how can anything be suppressed more than 100% (per the scale bar)? Is it possible to show the effect of CD on the same plot as the left column, so that the calculation of percent CD is less complicated to interpret?

Reviewer #3:

Skandalis et al investigate the role of corollary discharge and how it affects sensing external and self-generated stimuli in the zebrafish lateral line. They record from afferent neurons during stimulation of the lateral line and, by clustering, find four differently responding groups of neurons. By analyzing the different properties and modeling they claim that corollary discharge is an important feature in fast moving species as it prevents habituation of lateral line hair cells and therefore preserves sensitivity after swimming and prevents phase shifts. Unfortunately, this manuscript is extremely difficult to evaluate as the key findings are often insufficiently highlighted and the analyses are difficult to follow. It is therefore hard for this reader to share the manuscripts' conclusions. Most of the manuscript would benefit considerably from paragraphs with topic sentences and conclusions. Frankly, it is hard to discern what the major advance over work by Pichler and Lagnado 2019 and 2020 are; those data and conclusions — particularly about different subtypes of lateral line cells and heterogeneity — are highly similar and the Discussion here does not deal with that aspect. Finally, the authors are encouraged to look into recent work from John Tuthill's group and other colleagues examining the intersection of proprioceptive signals and movement; treatment of that literature in the Discussion would broaden the target audience.

Fig 1Bi should say "discharge" 

Fig 2 it's really not clear what "first" and "last" refer to — if the reader looks at the figures there's no chance they will understand what is going on without a deep dive into the text — maybe a visual indicator that "first/last" refer to the stimulus? 

Figure 2 Bi does not make a strong case for segregating groups 1&4 from one another. 

Figure 2 B what do a,b, and c refer to? Is there some relationship to the letters in Figure 3, which are similarly not described?

The abstract suggests that this is the main advance of the paper:

"Activation of the corollary discharge prevents habituation, reduces response heterogeneity, and regulates response phases in a narrow interval around the time of the peak stimulus."

But there is not clear evidence for these claims in Figure 3D — even given the manuscripts' segregation of afferent cell types. This is also decidedly frequency-dependent with very different consequences at 5 and 20/40Hz, suggesting that for the nervous system to make sense of this change in habituation it would also be dependent on the speed of the tail movement. The manuscript seems to agree with these points per 244-251 so why highlight these findings in the Abstract when they aren't central to the paper?

In Figure 3 C and D i-iv: what do the letters above the circles mean? 

Line 83: Because of which arrangement, precisely? 

Line 133: What, precisely, is meant by "vector strength?"

Line 219: What does "the smooth functions" refer to? 

Line 276-278: Why would you need CD (a signal that modulates with the swim command) to do that, rather than simply tonic inhibition during movement? 

Line 281: should be declines

Line 341: Given the claims in Figure 1, and the return at the end of the vesicular model to this point, it would be really useful to be explicit in precisely how much worse the "estimate of the body wave" would be as a function of CD strength. Absent that, it's really difficult to determine how important CD might be for the proposed computation it is supposedly influencing (eg line 361 "a vital role for CD.") Such a computation would constitute a meaningful step forward from the model/framing of Pichler and Lagnado.

Line 395-397: Could you explain a bit more what the function of Group 4 cells could be? 

Line 469: it seems like there is a part of the sentence missing

Reviewer #4:

This paper is centered on a very interesting phenomenon of high importance, namely understanding how sensory feedback from the mechanosensory lateral line contributes to the control of swimming in fishes. To do this they use an excellent model system, the larval zebrafish, which is amenable to electrophysiological recording in conjunction with lateral line stimulation. The Liao lab is a leader in this field and the technical expertise in these types of experiments is very strong. This is a question of high interest to the field of motor control, understanding proprioception's role in general, and it has been particularly poorly understood in fishes. Therefore, this work is very important and applicable beyond the zebrafish to sensory-motor integration in general.

There are several difficulties in the presentation of the manuscript which require minor revisions in my view. I am strongly motivated to understand this paper and yet I had great difficulty. Despite mostly clear introductory text, I found some general critiques or some components either missing or not clearly communicated for the non-expert.

1) The title of the paper starts with Corollary discharge (CD) yet the actual neurons responsible for providing this CD are not discussed for 1.5 pages, and then barely at all. What happens to swimming or to afferent firing if you remove CD entirely by some ablation or lesion? Has this been done? If so, please discuss. If not, can you do it here to test its role in your model?

2) There seems to be an assumption that a perfect internal representation of the body wave when swimming is essential, and I'm not sure why that needs to be the case. Please explain/expand.

3) From what I understood, in the experimental portion there were no measurements of CD, but simply the assumption that CD activity correlated with when swimming was occurring or not occurring as observed by the ventral root recordings. If that's the case, please make that clear and expand on that topic so as to justify to the non-expert reader why you made that assumption.

4) I did not understand Figure 3 at all. I am not a modeler but I'd like to be able to read the results and the figure and vaguely understand what is being done and what conclusion I should draw.

5) It was not clear to me which figures or figure panels represented computationally-generated data and which represented experimentally-generated data. Please make this abundantly clear throughout.

6) The figures are not simple. Some panels are busy and convoluted. There are often color codes, number codes, groups, symbols, shaded sections, and not all of them are clearly labeled or described in the legend. They feel too busy and demand too much of the reader. I did not feel that each panel presented one idea at a time and built logically panel by panel, but instead felt that each panel presented several overlapping ideas at a time, which made comprehension difficult. Also please revisit your color choices in Figure 2 to make them friendly to the red-green colorblind. I will list some feedback below.

7) The writing style is at times excessively convoluted particularly for the non-expert. Please clarify throughout when possible. 

In general, this is a really excellent question to be asking but I cannot judge how well the methods are suited to answer it because I could not follow the presentation, and I really wanted to, because this is interesting. Making the figures and results less opaque will greatly benefit the authors as it would allow the true impact of the findings to go beyond an audience both expert in the larval zebrafish lateral line AND in computational approaches, and to reach an audience interested in sensory-motor integration and proprioception generally.

Detailed points on the writing:

Line 115- rostro-caudal?

Line 129- Is 38 the correct reference here?

Line 168- Do you mean Figure 2?

Line 171-173- Is the verb missing or something I didn't get in this sentence?

Line 468-469- Is there a word missing?

Detailed points on the Figures:

Figure 1-

Ai- Do we need these? What are we gaining? If this is just to show how you measure the angle you can do that with a cartoon fish or stick fish and the angle like you show in the shaded box.

Aiii- What are the dots on the side? I'm assuming the black dots on the fish below represent position of neuromasts but not explicitly stated in legend.

B- This was generally confusing to me and took a while to get, even though I think the concept is very good. The line placement of "Corollary discharge OFF" for example had me thinking the top line was with CD and second line was with CD OFF- the labeling was just visually confusing. I assume the thick black or red vertical lines correspond to peak firing? This is not written in the legend. Theta is not defined apart from by the cartoon. Also what is "q"? B is a really critical set up figure and I think you should make it less complicated visually if you can. This is the time in text as well to explain/convince why your idealized version is actually necessary and that when CD is off this is detrimental.

Figure 2-

Ai- I would suggest moving the Afferent trace to the bottom so that it's under the VR and the stimulus? Just an idea. Would make it easier to understand I think and to compare to Aiii.

Why is the stimulus trace black and shaded with blue instead of just blue?

Why is the text for Ventral root in red if the trace is black?

If swim and CD ON are the same, and inactive and CD OFF are the same, as indicated in the legend, why not put Swim (CD ON) on the red sections rather than put a separate label on different parts of the plot?

What is the necessity of having a post-swim shading? This is not clear.

Take more space, I can hardly see the scale bars, etc. It's all rather small and scrunched compared to B.

Aii- In the supp fig it says red is swimming but this is not stated here.

Aiii - This is complex and the legend is not sufficient. What is the main bold line and how is it calculated? I think understanding Delta E and Delta S is important and it is not intuitive as is. I suggest you break this down clearly here so that 1) B will make sense and 2) the reader can internalize it because they understand it clearly. They need to learn this new metric and have it at their fingertips going forward so it needs to be obvious here.

B- Make color choices color-blind friendly. Not everyone can distinguish these 4 colors.

In Bi what are the large circles around each cluster? Are these calculated or approximated? No reference in the legend.

By B ii and B iii I was lost.

Figure 3, 4, 5- I was completely lost by here.

---

## [Decision Letter · Decision Letter 2]

31 Aug 2021

Dear Dimitri,

Thank you for submitting your revised Research Article entitled "The role of corollary discharge on proprioception from lateral line sensory feedback" for publication in PLOS Biology. I have now obtained advice from three of the the original reviewers and have discussed their comments with the Academic Editor. 

Based on the reviews, we will probably accept this manuscript for publication, provided you satisfactorily address the remaining points raised by the reviewers. Please also make sure to address the following data and other policy-related requests.

IMPORTANT:

a) Please address the remaining concerns from reviewers #2 and #4.

b) Please could you choose a more explicit and informative title? Please include an active verb and avoid punctuation. We suggest: "Corollary discharge helps to ensure precise proprioceptive feedback from the lateral line organ during undulatory locomotion in fish"

c) We note that in your funding statement you say “JCL: Private foundation grants (sources undisclosed)” - this is not permissible - all sources of funding must be disclosed. Please give full details.

d) Many thanks for depositing your data and code in Figshare. Please cite this data location in all relevant main and supplementary Figure legends, e.g. "The data and code underlying this Figure may be found at [Figshare URL]"

We expect to receive your revised manuscript within two weeks. 

*Published Peer Review History*

*Early Version*

Sincerely,

Roli

Senior Editor,

rroberts@plos.org,

PLOS Biology

DATA NOT SHOWN?

REVIEWERS' COMMENTS:

Reviewer #2:

The authors have significantly improved the presentation and clarity of their earlier manuscript in this revision. They have carried out a major rewrite and reorganization to address reviewers' concerns. I think that they have addressed all points effectively, and in particular Figures 3, 4, and 6 are tremendously easier to interpret now. The results continue to be of significant importance to the field. 

I have one minor suggestion at the authors' discretion, which would be to remove the paragraph at lines 305-322 (or place it elsewhere). I found it disrupted the flow of description of Figure 3, and I couldn't understand why it was in the middle of the Results section, which is typically a more straightforward figure-by-figure presentation. However this is an optional suggestion.

Reviewer #3:

The authors have addressed the concerns raised by the reviewers. The manuscript is a lot easier to read and understand and the figures are simpler and easier to interpret, also the additional Figure 1 helps a lot. 

Reviewer #4:

The authors have undertaken a SUBSTANTIAL revision here which has greatly improved the clarity and readability of the manuscript and should be applauded. I imagine this was a lot of work and as a reviewer I greatly appreciate that my review was taken seriously and implemented on such a scale. I have a few minor questions/comments/suggestions remaining.

1. In general, I feel like I understood a lot from reading the response to Reviewer #3 particularly regarding the relationship between your work and the previous recent Lagnado papers. That was valuable to me and I would encourage you to spell that out in the manuscript explicitly. That goes for a lot of what you wrote in the response to reviewers, I feel like many of those points should be in the manuscript.

2. Thank you for further clarifying and explaining the source of the CD and your assumptions around this. Although you state it has been extensively verified, it's still not clear to me why this signal could not be measured or manipulated directly, rather than inferred. Can you expand on this a bit? I'm not saying you should do it if it's technically challenging enough to be a whole other paper, but address it for the reader because it's not clear to me why that wouldn't be the perfect thing to do. Why not say optogenetically mess with the efferent activity? Is this possible? Wouldn't this add to your findings if these manipulations agree with your model? Just discuss please. Also I disagree that you can't do a laser ablation and then have a fish swim in a behavior or perform in a physiology experiment afterward because I've done that myself. It's annoying though, to be sure. I'm not saying you need to do it, but it's very possible and ablation + behavior experiments have been published since Liu and Fetcho 1999.

3. The figures are much much MUCH clearer but they are still a bit in a roundabout order (Figure 3A, Figure 4, Figure 5, Figure 6, Figure 3B, Figure 4, etc etc) so I suggest either discussing them in the order presented or presenting them in the order you mean to discuss them.

4. Re Figure 5, it wasn't clear to me why you only looked at post swim and not also pre swim.

5. My main issue with this paper is still the readability, even though it must be said it is DRASTICALLY improved from the previous version. I often found myself in the middle of a results paragraph saying "why are we doing this?" or at the end of one saying "what is the take home here?" I would recommend more concise guiding sentences for the reader. There is an absolute ton of complicated analysis here and while it's now possible for me to understand it (yay!) it's not always clear why this is being undertaken, what is being learned and gained, from each plot or figure. Perhaps that's just that this isn't my exact area of work (as you can tell) but I'd imagine you'd like this paper to appeal to folks like me. There are also still labels and descriptions missing on the figures and legends so please review carefully. If you're skipping axis labels for an aesthetic reason I would suggest against this as it was a bit confusing especially in cases like Figure 4. If the reader is less annoyed they will be more likely to read the whole thing.

---

## [Editor Report · Decision Letter 3]

21 Sep 2021

Dear Dimitri,

On behalf of my colleagues and the Academic Editor, Tom Baden, I'm pleased to say that we can in principle offer to publish your Research Article "Corollary discharge enables proprioception from lateral line sensory feedback" in PLOS Biology, provided you address any remaining formatting and reporting issues. These will be detailed in an email that will follow this letter and that you will usually receive within 2-3 business days, during which time no action is required from you. Please note that we will not be able to formally accept your manuscript and schedule it for publication until you have made the required changes.

PRESS: We frequently collaborate with press offices. If your institution or institutions have a press office, please notify them about your upcoming paper at this point, to enable them to help maximise its impact. If the press office is planning to promote your findings, we would be grateful if they could coordinate with biologypress@plos.org. If you have not yet opted out of the early version process, we ask that you notify us immediately of any press plans so that we may do so on your behalf.

Best wishes,

Roli 

Roland G Roberts, PhD 

Senior Editor 

PLOS Biology

rroberts@plos.org